# DENV NS1 and MMP-9 cooperate to induce vascular leakage by altering endothelial cell adhesion and tight junction

Pan Pan[1,2☉], Geng Li[1,3,4☉], Miaomiao Shen[3], Zhenyang Yu[3], Weiwei Ge[3], Zizhao Lao[4], Yaohua Fan[4], Keli Chen[3], Zhihao Ding[3], Wenbiao Wang[1], Pin Wan[1], Muhammad Adnan Shereen[3], Zhen Luo[1,5], Xulin Chen[1,5], Qiwei Zhang[1,5], Luping Lin[4,6]*, Jianguo Wu[1,2,3,5]*

1 Guangdong Provincial Key Laboratory of Virology, Institute of Medical Microbiology, Jinan University, Guangzhou, China, 2 The First Affiliated Hospital of Jinan University, Guangzhou, China, 3 State Key Laboratory of Virology, College of Life Sciences, Wuhan University, Wuhan, China, 4 Center for Animal Experiment, Guangzhou University of Chinese Medicine, Guangzhou, China, 5 Foshan Institute of Medical Microbiology, Foshan, China, 6 Guangzhou Eighth People's Hospital, Guangzhou, China

☉ These authors contributed equally to this work.
* linlupingdoctor@qq.com (LL); jwu898@jnu.edu.cn (JW)

**Data Availability Statement:** All relevant data are within the manuscript and its Supporting Information files.

## Abstract

Dengue virus (DENV) is a mosquito-borne pathogen that causes a spectrum of diseases including life-threatening dengue hemorrhagic fever (DHF) and dengue shock syndrome (DSS). Vascular leakage is a common clinical crisis in DHF/DSS patients and highly associated with increased endothelial permeability. The presence of vascular leakage causes hypotension, circulatory failure, and disseminated intravascular coagulation as the disease progresses of DHF/DSS patients, which can lead to the death of patients. However, the mechanisms by which DENV infection caused the vascular leakage are not fully understood. This study reveals a distinct mechanism by which DENV induces endothelial permeability and vascular leakage in human endothelial cells and mice tissues. We initially show that DENV2 promotes the matrix metalloproteinase-9 (MMP-9) expression and secretion in DHF patients' sera, peripheral blood mononuclear cells (PBMCs), and macrophages. This study further reveals that DENV non-structural protein 1 (NS1) induces MMP-9 expression through activating the nuclear factor κB (NF-κB) signaling pathway. Additionally, NS1 facilitates the MMP-9 enzymatic activity, which alters the adhesion and tight junction and vascular leakage in human endothelial cells and mouse tissues. Moreover, NS1 recruits MMP-9 to interact with β-catenin and Zona occludens protein-1/2 (ZO-1 and ZO-2) and to degrade the important adhesion and tight junction proteins, thereby inducing endothelial hyperpermeability and vascular leakage in human endothelial cells and mouse tissues. Thus, we reveal that DENV NS1 and MMP-9 cooperatively induce vascular leakage by impairing endothelial cell adhesion and tight junction, and suggest that MMP-9 may serve as a potential target for the treatment of hypovolemia in DSS/DHF patients.

**Funding:** This work was supported by the National Natural Science Foundation of China (81730061 to JW, 81973549 to GL, 81803813 to GL, and 81902053 to WB), Guangdong Province Introduction of Innovative R&D Team (2017ZT07Y580 to JW), Postdoctoral Research Foundation of China (2020M683177 to PP), GZUCM First-Class Universities and Top Disciplines Projects (2021XK16 to GL and), Open Research Fund Program of the State Key Laboratory of Virology of China (2021KF003 to PP), and Key-Area Research and Development Program of Guangdong Province (2020B1111100002 to GL). The funders had no role in study design, data collection and analysis, decision to publish, or preparation of the manuscript.

**Competing interests:** The authors have declared that no competing interests exist.

## Author summary

DENV is the most common mosquito-transmitted viral pathogen in humans. In general, DENV-infected patients are asymptomatic or have flu-like symptoms with fever and rash. However, in severe cases of DENV infection, the diseases may progress to dengue hemorrhagic fever (DHF) or dengue shock syndrome (DSS), the leading causes of morbidity and mortality in school-age children in tropical and subtropical regions. DENV-induced vascular leakage is characterized by enhanced vascular permeability without morphological damage to the capillary endothelium. This study reveals a possible mechanism by which DENV NS1 and MMP-9 cooperatively induce vascular leakage. NS1 also recruits MMP-9 to degrade β-catenin, ZO-1, and ZO-2 that leads to intervene endothelial hyperpermeability in human endothelial cells and mouse vascular. Moreover, the authors further reveal that DENV activates NF-κB signaling pathway to induce MMP-9 expression in patients, mice, PBMC, and macrophages though NS1 protein. This study would provide new in signs into the pathogenesis of DENV infection, and suggest that MMP-9 may act as a drug target for the prevention and treatment of DENV-associated diseases.

## Introduction

Dengue virus (DENV) is the most common mosquito-transmitted viral pathogen in humans. As reported by the World Health Organization (WHO), an estimated 40% of the world population is at risk of DENV infection, and approximately 390 million people worldwide are infected with DENV every year [1–3]. As mosquitoes are moving to new areas because of a climate change, the disease is spreading to less tropical and more temperate countries. WHO has named dengue as one of the world's top 10 threats to global health in 2019 [4]. In general, DENV-infected patients are asymptomatic or have flu-like symptoms with fever and rash. However, in severe cases of DENV infection, the disease may progress to dengue hemorrhagic fever (DHF) or dengue shock syndrome (DSS), the leading causes of morbidity and mortality in school-age children in tropical and subtropical regions [1, 5]. According to the latest WHO classification, dengue severity is divided into dengue without warning signs, dengue with warning signs, and severe dengue. Vascular leakage, as one of the key features of DHF/DSS and severe dengue, is closely associated with increased vascular permeability in DENV-infected patients [6]. The presence of vascular leakage causes hypotension, circulatory failure, and disseminated intravascular coagulation as the disease progresses, which can lead to the death of DHF/DSS patients. So far, there is no licensed antiviral treatment, but only has supportive therapy, including fluid management, available for patients with vascular hyperpermeability as the mechanism underlying the phenomenon remains unclear.

DENV-induced vascular leakage is characterized by enhanced vascular permeability without morphological damage to the capillary endothelium. Despite the findings on DENV replication in some human endothelial cells (ECs), results from the postmortem analysis of DENV-infected human tissues indicate no generalized DENV ECs infection, which is supported by the fact that patients with severe DENV infection manage to fully recover in a short time [7]. All evidence that the loss of vascular integrity and function in DENV infection *in vivo* is not caused by extensive damage to the endothelium. Instead, vasoactive endothelial factors released from DENV-infected cells appear to play a major role in this phenomenon. During DENV infection, viruses mainly target at monocytes/macrophages and dendritic cells (DCs) for replication *in vivo*. Changes in production of interleukin-1 (IL-1), interleukin-6 (IL-6), macrophage inhibitory factor (MIF), tumor necrosis factor α (TNF-α), and metalloproteinases

are noted in macrophages and DCs infected with DENV *in vitro* [8, 9]. A study of EC barrier function using *in vitro* models to describe movement of labeled macromolecules or changes in cell electrical resistance demonstrated that soluble factors released from DENV-infected macrophages could change the permeability of an EC monolayer in the absence of relevant viral-induced cytopathic effect [10]. Taken together, altered production of factors released from circulating monocytes, macrophages, and DCs in human tissues occurs upon DENV infection, and these factors may coordinate to induce functional changes in endothelial cells.

The maintenance of endothelial cell permeability is mainly determined by two factors. One is the polyglycoprotein complex that forms a protective membrane on the cell surface to ensure the integrity of endothelial cells [11, 12]. The other is the adhesion and tight junction between endothelial cells, which play important roles in maintaining the integrity of endothelial cells [13]. In addition to promoting cell adhesion, the adhesion and tight junctions can also regulate cell growth, apoptosis, gene expression, and cardiovascular formation by altering intracellular signals [13]. The adhesion and tight junction proteins play key roles in maintaining homeostasis. There are two major types of junction, adhesion junction and tight junction [13]. The changes in adhesion and tight junction structures are regulated by matrix metalloproteinases (MMPs), which are destructive to the integrity of endothelial cells [14]. In the MMP family, the MMP-9 protein promotes tumor migration by degrading extracellular matrix [15, 16]. Although previous study reported that DENV induces vascular leakage by up-regulating the expression of MMP-9 in Dendritic cells (DCs) [17], the specific mechanism underlying this regulation is not clear.

Endothelial glycocalyx has been shown to increase endothelial permeability through degradation. Under normal physiological conditions, the glycocalyx as a barrier controls a number of physiological processes, which particularly prevents leukocytes and platelets from adhering to vessel walls [18]. In addition, the degradation of the glycocalyx is closely related to severe vascular leakage in DENV infections. However, it is not fully understood the causes of glycocalyx degradation upon DENV infection. DENV non-structural protein 1 (NS1) is an established early diagnostic marker for DENV infection. The serum concentration of NS1 can reach up to 50 µg/ml in a DHF/DSS case, indicating that NS1 is positively correlated with the disease severity. DENV NS1-induced vascular leakage has been extensively discussed since 2015 [19]. A previous study reported that NS1 induced vascular leakage in mice and anti-NS1 antibodies played a role in reducing NS1-induced vascular leakage and the mortality rate. A study suggested that NS1 induced vascular leakage *via* Toll-like receptor 4 (TLR4) [20]. Another study reported that autophagy-mediated junction disruption was associated with DENV NS1-induced vascular leakage, which may explain why vascular leakage in DENV-infected patients is a rapid and reversible pathogenic change [21]. NS1-induced MIF secretion is involved in NS1-induced EC autophagy. In addition, an *in vitro* study showed that DENV-infected cells increased endothelial permeability by inducing MIF secretion. NS1 not only disrupts endothelial junctions but also causes vascular leakage through HPA-1-mediated glycocalyx degradation. In short, there is growing evidence indicating that NS1 plays a critical role in dengue pathogenesis as it causes vascular leakage and hemorrhage during DENV infection [22]. Vascular permeability changes can be induced by destroying the thin (about 500 nm) and gel-like endothelial glycocalyx layer (EGL) that coats the luminal surface of blood vessels [23, 24]. Although vascular permeability changes are the main research focus, the specific mechanism underlying DENV pathogenesis needs to be further investigated.

In the present study, we reveal a distinct mechanism by which DENV induces endothelial permeability and vascular leakage in human endothelial cells and mouse tissues. DENV2 infection induces MMP-9 expression and secretion in human peripheral blood mononuclear cells (PBMCs) and macrophages through NS1-induced activation of the NF-κB signaling pathway.

More interestingly, NS1 also interacts with MMP-9, resulting in the degradation of important adhesion and tight junction proteins, impairing the adhesion and tight junctions, and consequently inducing endothelial hyperpermeability and increasing vascular leakage in human endothelial cells and mouse tissues. Collectively, these findings demonstrate that NS1 and MMP-9 cooperate to cause endothelial hyperpermeability and vascular leakage by impairing endothelial cell adhesion and tight junctions.

## Results

### DENV enhances MMP9 production in severe dengue patients

Previous studies have found that NS1 protein produced during dengue virus infection closely correlated with the onset of disease by promoting vascular leakage. In our study, we found that the concentrations of NS1 protein in the serum samples of severe dengue patients continued increase with the prolongation of the infection time (Fig 1A and 1B). Previous studies have reported that matrix metalloprotein-9 (MMP-9, also known as Gelatinase B, GelB) produced in DENV-infected dendritic cells could induce vascular leakage. We also noticed that the concentrations of MMP-9 protein in the serum samples of severe dengue patients increased over the course of DENV infection (Fig 1C and 1D). Statistical analysis showed a close correlation

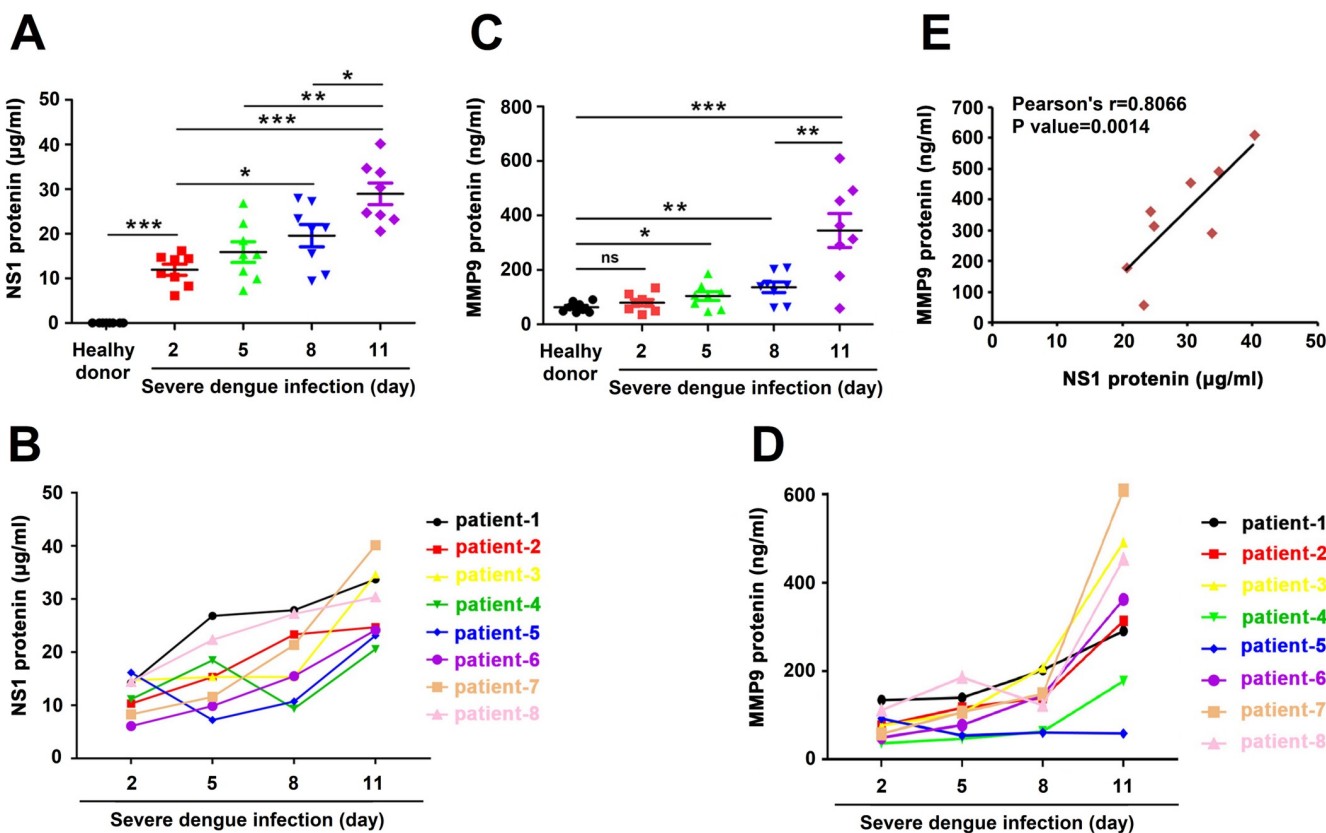

**Fig 1. DENV induces the production of NS1 and MMP-9 in severe dengue patients.** (**A** and **B**) The serum concentrations of NS1(A) and MMP-9 (B) in healthy donors and severe dengue patients infected days (2, 5, 8, and 11 days) were measured by ELISA. Points represent the value in each serum sample. (**C** and **D**) The serum concentrations of NS1 (C) and MMP-9 (D) in each severe dengue patients infected days (2, 5, 8, and 11 days) were measured by ELISA. Points represent the value in each serum sample. (**E**) the correlations of the concentrations of NS1 and MMP-9 in the same group of severe dengue patients infected 11 days were plotted. Linear regressions were traced according to the distributions of the points. Dates were representative of two independent experiments. ns means not significant. P ≤ 0.05 (*), P ≤ 0.01 (**), P ≤ 0.001 (***).

between NS1 and MMP-9 (Fig 1E). These results suggest that MMP9 production is enhanced in severe dengue patients and is correlated with NS1 production.

## DENV NS1 interacts with MMP-9

To assess DENV protein(s) involved in the regulation of MMP-9 production, we determined the interactions of MMP-9 with the viral proteins. Human embryonic kidney 293T (HEK293T) cells were co-transfected with pHA-MMP-9 and each plasmid (pFlag-Cap, pFlag-M, pFlag-Prm, pFlag-E, pFlag-NS1, pFlag-NS2A, pFlag-NS2B, pFlag-NS3, pFlag-NS4A, and pFlag-NS4B) expressing individual DENV proteins (Cap, M, Prm, E, NS1, NS2A, NS2B, NS3, NS4A, and NS4B), respectively. Co-immunoprecipitation (Co-IP) assays showed that only DENV NS1 and NS3 proteins interacted with MMP-9 (Fig 2A). In this study, we primarily focused on NS1 but not NS3 because both MMP-9 and NS1 are secreted proteins [25] and NS1 plays a role in the regulation of vascular leakage upon DENV infection [19]. Co-IP analysis confirmed that MMP-9 interacted with NS1 (Fig 2B). Western-blot results showed that MMP-9, NS1, E, and NS4A proteins were expressed and detected in both HEK293T cells (S1A Fig) and Hela cells (S1B Fig). BiFC assays further demonstrated that NS1 protein interacted with MMP-9 protein, but DENV E protein and NS4A protein could not interact with MMP-9 protein, in both HEK293T cells (Fig 2C and 2D) and Hela cells (S1C and S1D Fig). Notably, His-pull-down assays demonstrated that purified DENV2 NS1 protein could directly interact with purified MMP-9 protein, but purified SARS-CoV-2 N protein failed to interact with purified MMP-9 protein (Fig 2E), indicating that DENV2 NS1 protein specifically and directly interact with MMP-9 protein. Yeast-two hybridization assays also showed that NS1 and MMP-9 could interact with each other (Fig 2F), but E or NS4A failed to interact with MMP-9 (S1E Fig). We showed that MMP-9, NS1, E, and NS4A proteins were expressed and detected in the yeast cells (S1F Fig). Moreover, the domains of MMP-9 involved in the MMP-9/NS1 interaction were determined by progressive truncation of MMP-9 (D1-D9) (Fig 2G). HEK293T cells were co-transfected with pHA-NS1 along with each of the plasmids expressing truncated MMP-9 protein, respectively. Co-IP results showed NS1 interacted with MMP-9 D1 (106–707), MMP-9 D4 (106–511), MMP-9 D5 (1–511), MMP-9 D6 (106–440), MMP-9 D7 (106–233), MMP-9 D8 (106–397), and MMP-9 D9 (223–440), but not with MMP-9 D2 (440–707) or MMP-9 D3 (512–707), suggesting that the Zinc-binding catalytic domain and the Fibronectin type-like domain D6 (106–440) are involved in the MMP-9/NS1 interaction (Fig 2H). Taken together, the results demonstrate that DENV NS1 directly interacts with MMP-9.

## NS1 induces expression and proteolytic activity of MMP-9

The molecular mechanism by which NS1 regulates MMP-9 expression was investigated. HEK293T cells, Hela cells, THP-1 differentiated macrophages, and human umbilical vein endothelial cells (HUVECs) were transfected with plasmid pHA-NS1 at different concentrations. ELISA results indicated that the levels of NS1 protein in the supernatants were increased in pHA-NS1-concentration dependent fashions (S2A–S2D Fig) and similarly, Western-blot results showed that the levels of NS1 protein in the cell lysates were increased in pHA-NS1 concentration-dependent manners (S2E–S2H Fig), demonstrating that NS1 protein is secreted in the supernatants and expressed in the cell of these cell lines. Next, THP-1 differentiated macrophages were transfected with pFlag-NS1 at varying amounts. MMP-9 protein production (Fig 3A, top), MMP-9 enzyme activity (Fig 3A, middle), and MMP-9 mRNA transcription (Fig 3A, bottom) were enhanced by NS1 in dose-dependent manners. However, MMP-9 enzyme activity and MMP-9 protein production were not influenced by DENV-E or DENV-NS3 in THP-1 differentiated macrophages (S2I and S2J Fig). We noticed that unlike

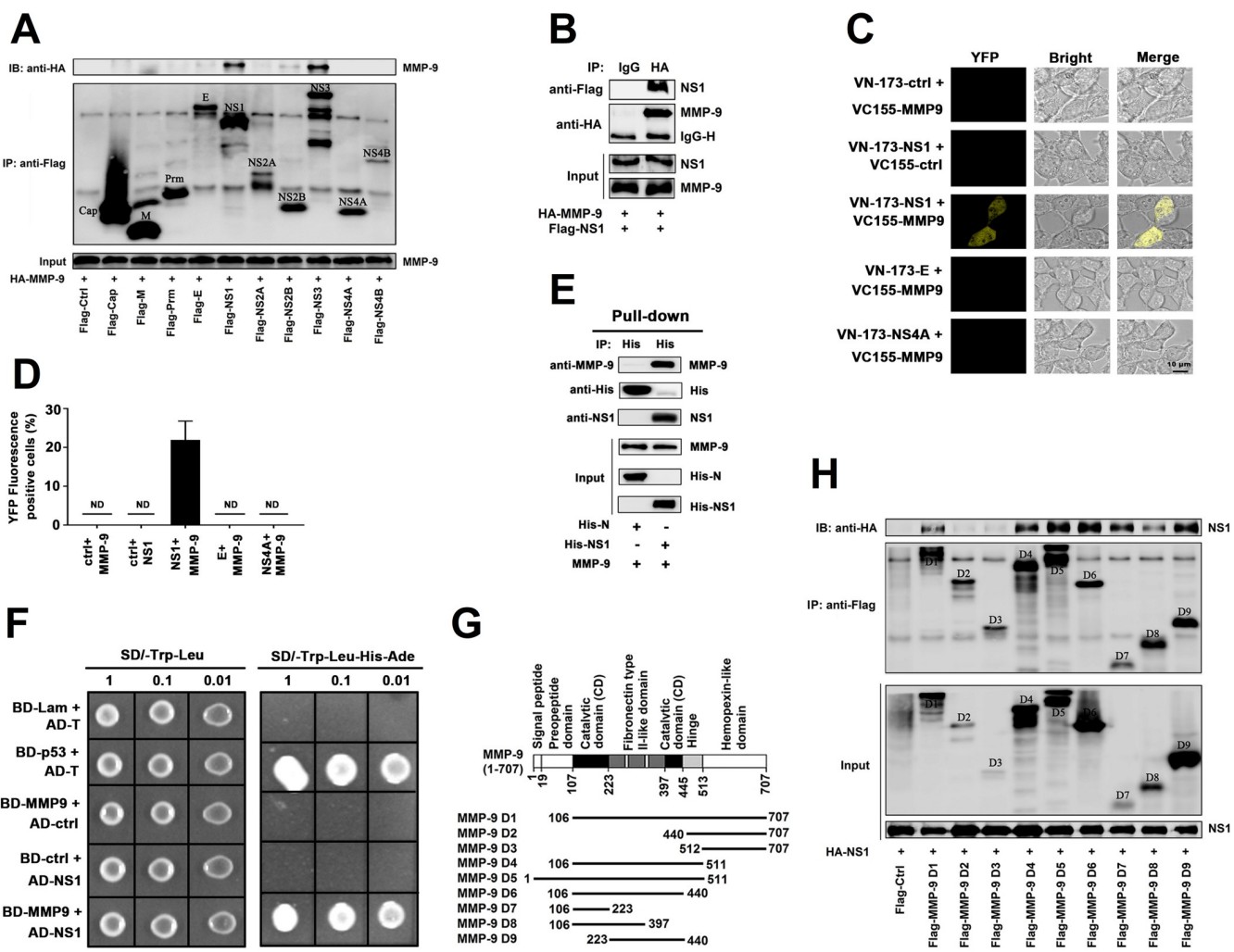

**Fig 2. DENV NS1 interacts with MMP-9.** (**A**) HEK293T cells were co-transfected with HA-MMP-9 and Flag-Cap, Flag-M, Flag-Prm, Flag-E, Flag-NS1, Flag-NS2A, Flag-NS2B, Flag-NS3, Flag-NS4A, or Flag-NS4B. Cell lysates were immunoprecipitataed using anti-Flag antibody, and analyzed using anti-Flag and anti-HA antibody. Cell lysates (40 μg) was used as Input. (**B**) HEK293T cells were co-transfected with HA-MMP9 and Flag-NS1, Cell lysates were immunoprecipitataed using anti-HA antibody, and analyzed using anti-Flag and anti-HA antibody. Cell lysates (40 μg) was used as Input. (**C, D**) HEK293T cells were co-transfected with empty vector or VC-155-MMP-9 and VN173-NS1/E/NS4A. At 24 h post-transfection, living cells were observed by confocal microscopy. The quantification of YFP-positive cells was determined by ImageJ software (D). ND means not detection. (**E**) Purified His-SARS-CoV-2-N (5 μg) or His-DENV2-NS1 (5 μg) was incubated with purified no-tagged MMP-9 protein (3 μg) for 24 h, Mixtures were incubated with Ni-NTA Agarose beads. Mixtures were analyzed by immunoblotting using anti-MMP9, anti-NS1, anti-His antibody. Untreated protein including His-SARS-CoV-2-N (1 μg), His-DENV2-NS1 (1 μg), or no-tagged MMP-9 protein (1 μg) were analyzed by immunoblotting using anti-MMP9, anti-NS1, and anti-His antibody (as input). (**F**) Yeast strain AH109 were co-transformed with combination of binding domain (BD-p53, BD-MMP-9, and BD-Lam) and activation domain (AD-T, AD-NS1) plasmid. Transfected yeast cells were grown on SD-minus Trp/Leu double dropout plates, and colonies were replicated on to SD-minus Trp/Leu/Ade/His fourth dropout plates to check for the expression of reporter genes. (**G, H**) Schematic diagram of wild-type MMP-9 protein and truncated mutants MMP-9 protein (D1 to D9) (G). HEK293T cells were co-transfected with HA-NS1 and Flag-MMP-9 truncated mutants (D1 to D9). Cell lysates were immunoprecipitated using anti-Flag antibody, and analyzed using anti-Flag and anti-HA antibody. Cell lysates (40 μg) was used as Input (H). Dates were representative of three independent experiments.

NS1 protein, the DENV NS3 and E proteins had no effect on the regulation of MMP-9 enzyme activity and MMP-9 protein production (S2K Fig). MMP-9 protein production and MMP-9 enzyme activity were also facilitated by NS1 in dose-dependent manners in HEK293T cells (Fig 3B). We also noticed that level of MMP-9 protein was induced by NS1 in THP-1 differentiated macrophages (S2G Fig) but relatively unaffected by NS1 protein in HUVECs (S2H Fig). We speculated that one of the reasons for this phenomenal is the expression level of MMP-9 in

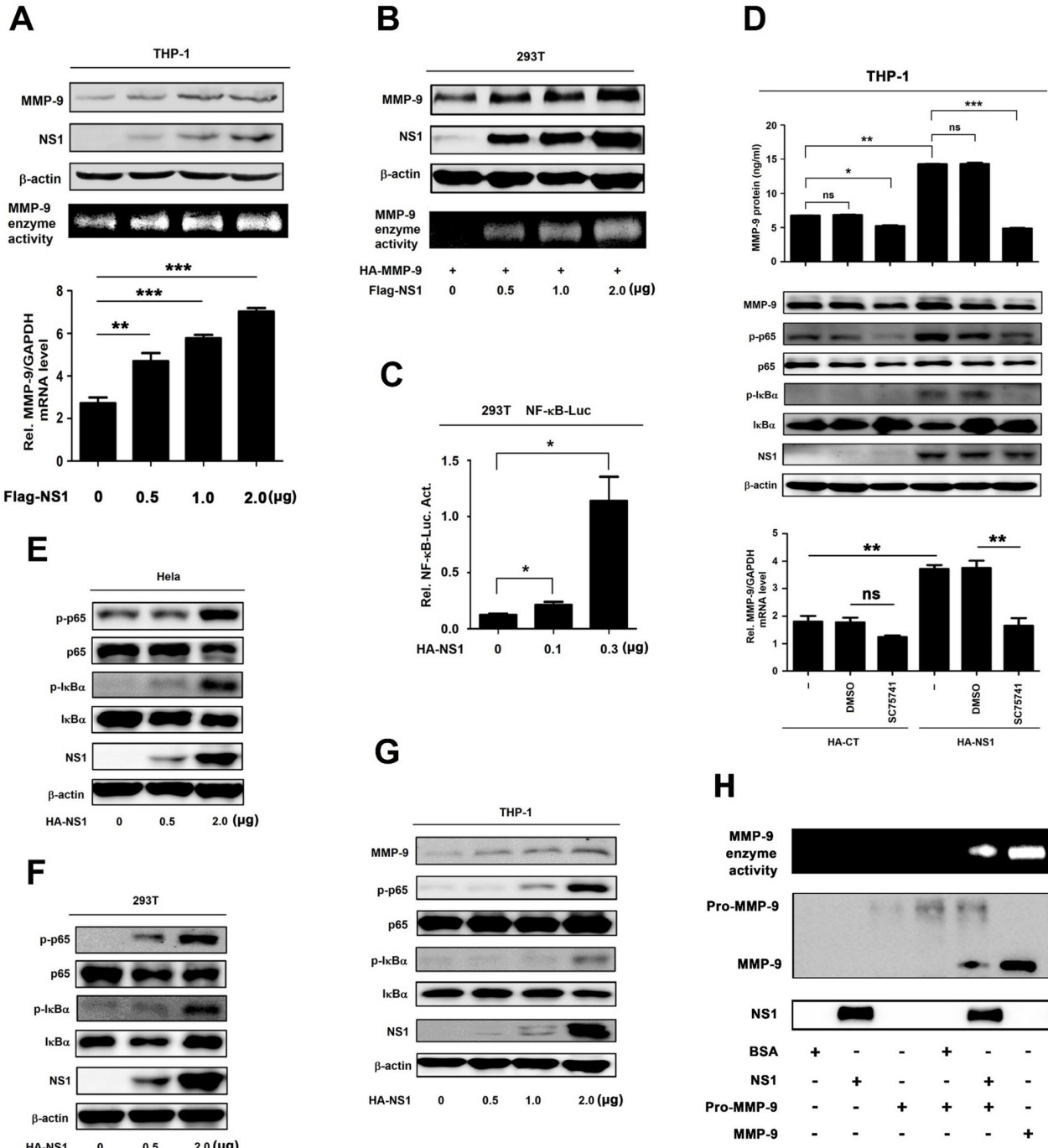

**Fig 3. NS1 induces expression and proteolytic activity of MMP-9.** (**A**) PMA-differentiated THP-1 macrophages were transfected with the different concentrations of plasmid encoding NS1 for 24 h. Cell lysates were analyzed (top) by immunoblotting. Supernatants were analyzed (middle) by gelatin zymography assays for MMP-9 proteinase activity. Intracellular MMP-9 RNA (bottom) was determined by qRT-PCR analysis. (**B**) HEK293T cells were co-transfected with the plasmid encoding MMP-9 and different concentrations of plasmid encoding NS1 for 24 h. Cell lysates were analyzed (top) by immunoblotting. Supernatants were analyzed (middle) by gelatin zymography assays for MMP-9 proteinase activity. Intracellular MMP-9 RNA (bottom) was determined by qRT-PCR analysis. (**C**) HEK293T cells were con-transfected with different concentrations of NS1 expressing plasmid and NF-κB reporter plasmid. Luciferase assays were performed 20 h after transfection. (**D**) PMA-differentiated THP-1 macrophages were firstly transfected with plasmid encoding HA-CT or HA-NS1 for 20 h, and then treated with 200 nM SC75741 for 5 h, MMP-9 protein in cell supernatants were measured by ELISA (top) and indicated proteins in cell extract were analyzed by WB (middle). Intracellular MMP-9 RNA (bottom) was determined by qRT-PCR analysis. (**E–G**) Hela cells (E), HEK293T cells (F), and PMA-differentiated THP-1 macrophages (G) were transfected with different concentrations of plasmid encoding NS1 for 24 h. The indicated proteins in cell extract were analyzed by WB. (**H**)

The supernatants of HUVEC cells were incubated with BSA (3 µg/ml), NS1 (3 µg/ml), pro-MMP-9 protein (200 ng/ml), BSA (3 µg/ml) plus pro-MMP-9 protein (200 ng/ml), NS1 (3 µg/ml) plus pro-MMP-9 protein (200 ng/ml) or MMP-9 protein (200 ng/ml) for 6 h, and then the supernatants were analyzed by gelatin zymography assays for MMP-9 proteinase activity and the indicated protein expression were analyzed by immunoblotting. Dates were representative of two to three independent experiments. ns means not significant. Values are mean ± SEM, P ≤ 0.05 (*), P ≤ 0.01 (**), P ≤ 0.001 (***).

HUVECs was much lower than that in THP-1 (S2G and S2H Fig). Taken together, the results suggest that DENV NS1 activates MMP-9 production, secretion, and enzyme activity.

It was reported that MMP-9 promoter contains the nuclear factor κB (NF-κB) regulatory elements [16]. Here, the role of the NF-κB binding sequences of MMP-9 promoter in the regulation of NS1-activated MMP-9 expression was determined. HEK293T cells were co-transfected with NF-κB reporter plasmid and pHA-NS1. Luciferase assays showed that NF-κB-Luc activity was significantly induced by NS1 in HEK293T cells (Fig 3C). Additionally, THP-1 differentiated macrophages were transfected with pHA-NS1 and treated with a specific inhibitor of NF-κB (SC75741). The results revealed that the levels of secreted MMP-9 protein in the cell culture supernatants (Fig 3D, top), MMP-9 protein in the cell lysates (Fig 3D, middle), and MMP-9 mRNA (Fig 3D, bottom) were down-regulated by SC75741, suggesting that NF-κB is required for NS1-induced production of MMP-9. Moreover, phosphorylated p65 (p-p65) and phosphorylated IκBα (p-IκBα) were induced by NS1 in Hela cells (Fig 3E), HEK293T cells (Fig 3F), and THP-1 differentiated macrophages (Fig 3G).

MMPs are secreted as inactive proenzymes and are activated subsequently by the cleavage of the propeptide domain by proteolytic enzymes. Results from gelatinase activity assays showed that gelatinase activity was only detected in the presence of both NS1 protein and pro-MMP9 protein, but the enzyme activity was not detected in the presence of BSA protein, NS1 protein, or pro-MMP9 protein alone, or in the presence of BSA and pro-MMP9 (Fig 3H), demonstrating that NS1 regulates the protease activity of MMP9 by interacting with MMP9 to promote the maturation of pro-MMP-9. Collectively, these results suggest that NS1 promotes MMP-9 expression through activating the NF-κB signaling pathway and modulates the proteolytic activity through interacting with MMP-9.

## DENV2 induces MMP-9 expression and secretion in human PBMCs and macrophages but not in HUVECs

MMP-9 is mainly produced in leukocytes and DENV infects leukocytes [26, 27]. Thus, we initially focused the role of DENV in the regulation of MMP-9 in human peripheral blood mononuclear cells (PBMCs). PBMCs isolated from healthy individuals were infected with DENV2 for different times or at different concentrations. Quantitative RT-PCR (qRT-PCR) showed that MMP-9 mRNA was induced upon DENV2 infection in PBMCs in a time- (S3A Fig, top) and dose-dependent fashion (S3B Fig, top). Gelatin zymography assays revealed that MMP-9 enzyme activity was enhanced by DENV2 infection in PBMCs (S3A and S3B Fig, middle). Results from qRT-PCR quantification indicate that DENV E gene mRNA was increased during virus infection (S3A and S3B Fig, bottom). Phorbol 12-myristate 13-acetate (PMA)-differentiated THP-1 macrophages (Fig 4A, top) and HUVECs (Fig 4A, bottom) were infected with DENV2 and UV-irradiated DENV2 for 48 h. Viral E mRNA was only detected in DENV2 infected cells, but not detected in the uninfected cells, the supernatant of C6/36 cells used as NO RT control, or UV-irradiated DENV2 treated cell (Fig 4A). Additionally, the role of DENV in the regulation of MMP-9 was determined in human acute monocytic leukemia cell line (THP-1). Phorbol 12-myristate 13-acetate (PMA)-differentiated THP-1 macrophages for different times or at different inoculum. Similarly, MMP-9 mRNA was up-regulated upon

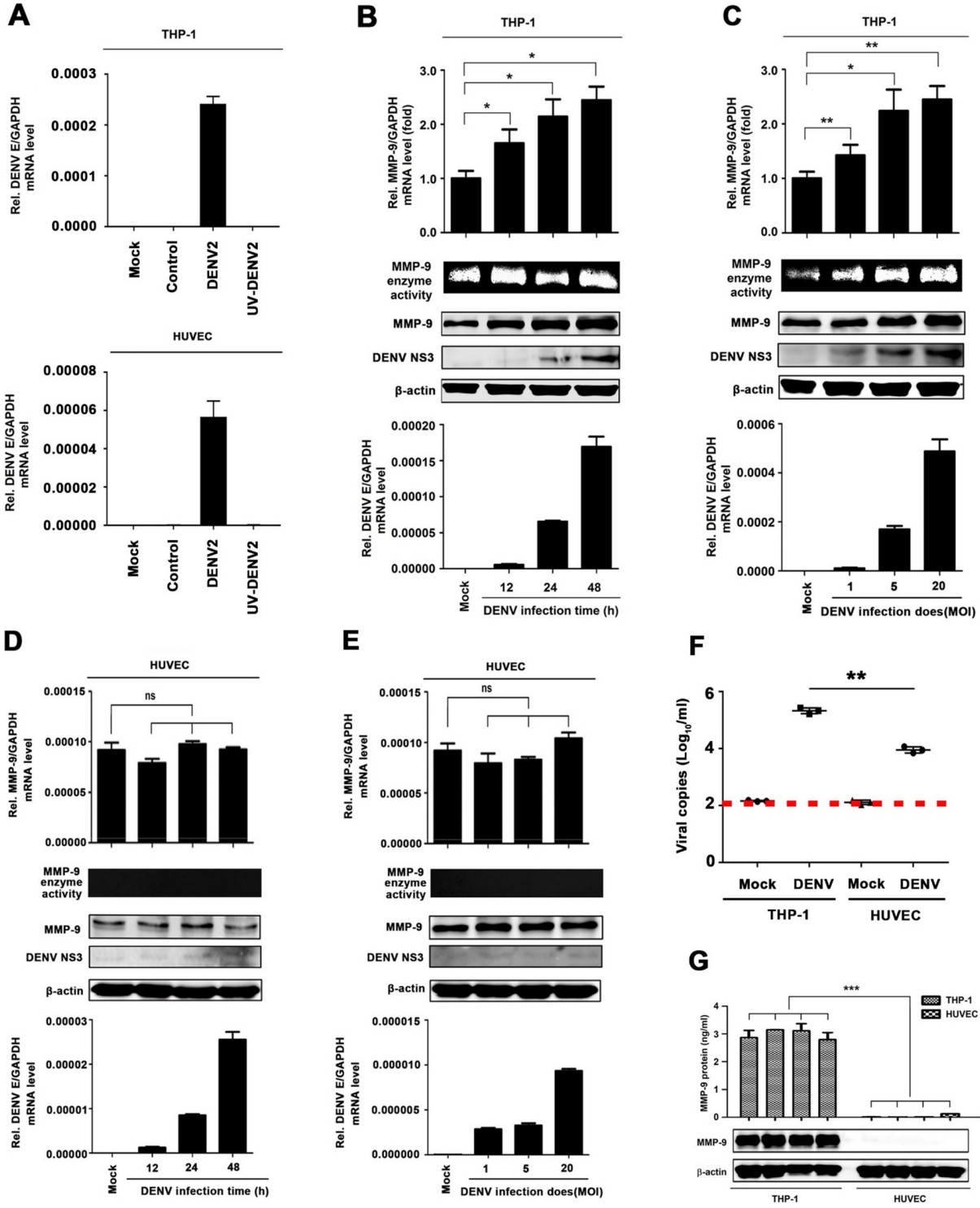

**Fig 4. DENV2 induces MMP-9 expression and secretion in human PBMCs and macrophages, but not in HUVECs. (A)** PMA-differentiated THP-1 macrophages (top) or HUVEC cells (bottom) were treated with infectious or UV- inactivated DENV2 at MOI = 5 for 48h. Intracellular DENV2 E RNA (bottom) was determined by qRT-PCR analysis. Mock: untreated cells. Control: supernatant of C6/36 cells without DENV2 infection. **(B and C)** PMA-differentiated THP-1 macrophages were infected with DENV2 for different times at MOI = 5 (B) or at different MOI for 24 h (C). Intracellular MMP-9 RNA (top) and DENV2 E RNA (bottom) was determined by qRT-PCR analysis, MMP-9 proteinase activity in the supernatants was determined by gelatin zymography assays and proteins in cell extract (middle) were analyzed by Western blotting. **(D and E)** HUVEC cells were infected with DENV2 for different times at MOI = 5 (DC) and at different MOI for 24 h (E).

Intracellular MMP-9 RNA (top) and DENV2 E RNA (bottom) was determined by qRT-PCR analysis, MMP-9 proteinase activity in the supernatants was determined by gelatin zymography assays and proteins in cell extract (middle) were analyzed by Western blotting. (**F**) HUVEC cells or PMA-differentiated THP-1 macrophages were infected with DENV2 at MOI = 5 for 24 h. Viral copies were quantified by RT-PCR. (**G**) HUVEC cells or PMA-differentiated THP-1 macrophages were equally distributed to four 12-hole plates and infected with DENV2 at MOI = 5 for 24 h. MMP-9 protein in cell supernatants were measured by ELISA (top) and indicated proteins in cell extract were analyzed by WB (bottom). Dates were representative of three independent experiments. ns means not significant. Values are mean ± SEM, P $\leq$0.05 (*), P $\leq$0.01 (**), P $\leq$0.001 (***).

DENV2 infection in DENV-infected THP-1 macrophages in a time- (Fig 4B, top) and dose-dependent fashions (Fig 4C, top). MMP-9 enzyme activity was also enhanced upon DENV2 infection (Fig 4B and 4C, middle). Viral E mRNA increased in proportion to infection time and inoculum (Fig 4B and 4C, bottom). Moreover, the levels of MMP-9 protein and DENV2 NS3 protein increased over the course of DENV2 infection (Fig 4B, middle) and correlated with virus inoculum (Fig 4C, middle). Interestingly, the levels of MMP-9 mRNA (Fig 4D and 4E, top), MMP-9 enzyme activity, and MMP-9 protein (Fig 4D and 4E, middle) remained unchanged in endothelial HUVECs upon DENV infection (Fig 4D and 4E, bottom). The ability of DENV replication was initially analyzed and compared in macrophages and endothelial cells. DENV2 viral copies was significantly higher in infected THP-1 differentiated macrophages as compared to that of human umbilical vein endothelial cells (HUVECs) (Fig 4F). Additionally, secreted MMP-9 protein increased in the supernatants of infected THP-1 differentiated macrophages but was undetectable in the supernatants of infected endothelial HUVECs (Fig 4G, top). Similarly, MMP-9 protein was highly expressed in the infected THP-1 differentiated macrophages, while modestly elevated in the infected HUVECs (Fig 4G, bottom). Taken together, these results indicate that DENV infection induced MMP-9 secretion in THP-1 differentiated macrophages but not in endothelial cells.

## NS1 promotes MMP-9-mediated endothelial hyperpermeability in human cells and mouse tissues

The biological effect of NS1 and MMP-9 in the regulation of endothelial cell permeability was evaluated. Firstly, the role of MMP-9 in the induction of endothelial cell permeability was determined. PMA-differentiated THP-1 macrophages were infected with DENV2 for different times, as indicated. The results showed that the secretion of NS1 in supernatants (Fig 5A, top) the production of MMP-9 protein and NS1 protein in cell lysates (Fig 5A, bottom) were up-regulated in time-dependent fashions. HUVECs grown on polycarbonate membrane system were incubated with the supernatants of DENV2-infected HUVEC cells or DENV2-infected THP-1 differentiated macrophages or pre-incubated with SB-3CT (a specific inhibitor of MMP-9) or irrelevant chemical inhibitor SC75741 (a specific inhibitor of NF-κB). Endothelial permeability was evaluated by measuring trans-endothelial electrical resistance (TEER) (ohm) using EVOM2 epithelial volt ohm meter. The level of TEER was not affected by the supernatants of DENV-infected HUVECs; significantly attenuated by the supernatants of DENV-infected THP-1 differentiated macrophages from 3 h to 15 h post-treatment; and however, such reduction was recovered by the treatment of SB-3CT but not recovered by the treatment of SC75741 (Fig 5B); suggesting that MMP-9 plays an important role in the induction of endothelial hyperpermeability mediated by DENV infection. Next, we further determined whether MMP-9 plays an important role in the induction of vascular permeability in mice after DENV infection. *IFNAR*[-/-] C57BL/6 mice were treated with PBS as a control group (n = 4), infected with DENV2 (n = 6), and intravenously treated with MMP-9 specific inhibitor SB-3CT and then infected with DENV2(NGC) (n = 6). DENV2 E and NS5 RNA were detected at high levels in the blood of DENV2-infected mice or SB-3CT-treated and DENV2-infected mice at 2 days

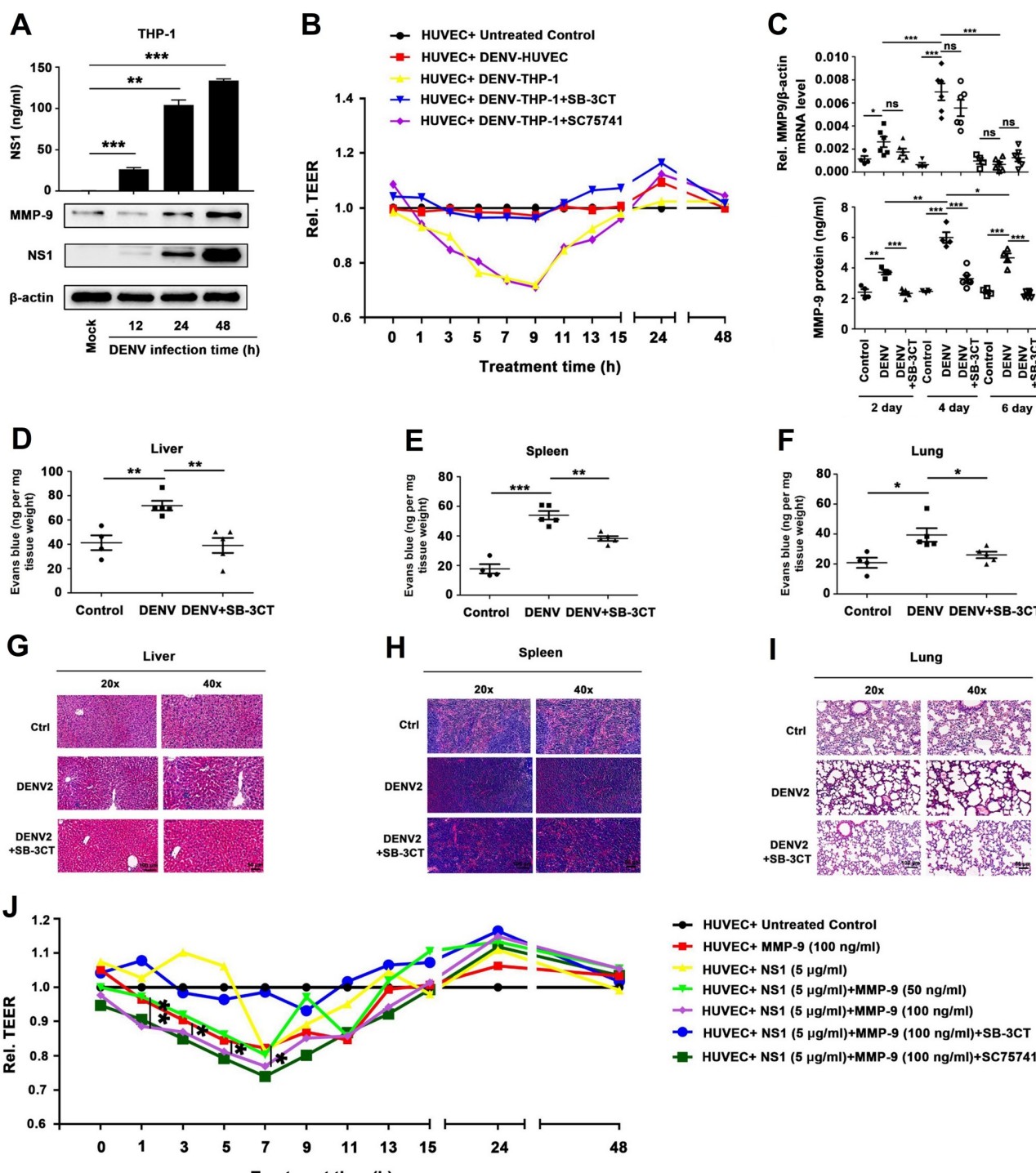

**Fig 5. NS1 facilitates MMP-9 to induce endothelial hyperpermeability in human cells and mice tissues.** (**A**) PMA-differentiated THP-1 macrophages were infected with DENV2 for different times at MOI = 5, NS1 protein in Supernatants were analyzed by ELISA (top). Cell lysates were analyzed by immunoblotting (bottom). (**B**) Confluent monolayers of HUVEC cells were grown on polycarbonate membrane system and treated with the supernatants came from DENV2 infected HUVEC cells or THP-1 cells for 24 h or pre-incubated with 600nM SB-3CT (a specific inhibitor of MMP-9 protein) or 600 nM SC75741 for 1h. Endothelial permeability was evaluated by measuring trans-endothelial electrical resistance (TEER) (ohm) using EVOM2 epithelial voltohmmeter. (**C–I**) IFNAR[-/-] C57BL/6 mice were intravenously injected with 300 μl DENV2 at a dose of $1\times10^6$ PFU/mouse (n = 6), pre-treated with 300 μl PBS containing MMP-9 specific inhibitor SB-3CT (5 mg/kg per mice) by intraperitoneal injection for 90 min and then treated with DENV2 ($1\times10^6$ PFU/mouse), repeat treated with SB-3CT (5 mg/kg per mice) on the fourth day after DENV2 (NGC) infection (n = 6), or 300 μl PBS containing the same volume DMSO as a control group (n = 4). 7 days after infection, mice were euthanasia, and the tissues were collected. MMP-9 RNA in the blood was

determined by qRT-PCR (upper) and MMP-9 protein in the serum was measured by ELISA (lower). Points represent the value of each serum samples (C). Evans blue dye was intravenously injected into mice 7 days after DENV infected groups (n = 5), control groups (n = 4) and DENV+SB-3CT (n = 5) (C–E). The dye was allowed to circulate for 2 hours before mice were euthanasia, tissues include liver (D), spleen (E) and lung (F) were collected, and the value of Evans blue was measured at $OD_{610}$. Histopathology analysis of tissues includes Liver (G), Spleen (H) and Lung (I) after DENV infection. (**J**) Monolayers of HUVEC cells grown on Transwell inserts were incubated for 48 h with MMP-9 protein (100 ng/ml) or NS1 protein (5 μg/ml) or NS1 (5 μg/ml) plus different concentration of MMP-9 (50 ng/ml to 100ng/ml) or pre-treated with 600 nM SB-3CT or 600 nM SC7574 for 1 h, then incubated with NS1 plus MMP-9. The TEER (ohm) was measured at indicated time points. Dates were representative of two to three independent experiments. ns means not significant. Values are mean ± SEM, P ≤0.05 (*), P ≤0.01 (**), P ≤0.001 (***).

and 4 days post-infection, but not detected in the blood of mocked-infected mice indicating that DENV2 replicated well in the mice (S4A and S4B Fig). We also noted that compared with DENV2-infected mice, the DENV2 E and NS5 RNA has no influence in SB-3CT-treated and DENV2-infected mice, indicating that SB-3CT had no effect on the replication of virus in mice (S4A and S4B Fig). It is worth noting that MMP-9 protein was significantly induced in the blood of DENV-infected mice, but not induced in the blood of mock-infected mice or SB-3CT-treated and DENV-infected both at 2 days, 4 days and 6 days post-treatment (Fig 5C, bottom); MMP9 mRNA was significantly induced in the blood of DENV-infected or SB-3CT-treated and DENV-infected, but not induced in the blood of mock-infected mice at 2 days and 4 days post-treatment (Fig 5C, top). Moreover, the intensities of Evans blue dye in the Liver, Spleen and Lung of DENV-infected mice were significantly higher than that mock-infected mice or SB-3CT-treated and DENV-infected mice tissues (Fig 5D–5F), suggesting that DENV infection induces vascular leakage in mice through promoting MMP-9 production. Meanwhile, Histopathology analysis showed that tissue injury like the spaces between the cells of tissue became larger in the Liver and Lung, the boundary between red pulp and white pulp were disrupted and the lymphatic nodules and pulping cells were increased in spleen were induced in DENV-infected mice organs compared with mock-infected mice tissues, but this phenomenon was rescued in SB-3CT-treated and DENV-infected mice tissues (Fig 5G–5I). Additionally, HUVECs grown on Transwell inserts were incubated with purified MMP-9 protein, NS1 protein, and NS1 protein plus MMP-9 protein at different concentrations, or pre-treated with SB-3CT or irrelevant chemical inhibitor SC75741 and then incubated with NS1 protein plus MMP-9 protein. The level of TEER was reduced by MMP-9 protein alone, NS1 protein alone, and MMP-9 protein plus NS1 protein from 2 to 7 h post-treatment; but the reductions were recovered by SB-3CT and not restored by SC75741 (Fig 5J); demonstrating that MMP-9 induces endothelial hyperpermeability in human endothelial cells.

## NS1 recruits MMP-9 to disrupts the junctions between endothelial cells

Changes in endothelial cell permeability can be achieved by destroying the endothelial glycocalyx layer (EGL) on the surface of endothelial cells or by altering the adhesion and tight junctions between endothelial cells [12, 13, 28, 29]. Previous study reported that DENV NS1 disrupts the EGL, leading to hyperpermeability [23]. However, the roles of NS1 in the regulation of the adhesion and tight junctions between endothelial cells have not been reported. Here, the expression of junction molecular include E-cadherin, ZO-2, α-E-catenin, and β-catenin was detected. The results showed that E-cadherin, β-catenin, and ZO-2 proteins were down-regulated by DENV infection compared with mock-infected mice in the liver (Fig 6A), spleen (Fig 6B), and lung (Fig 6C), but rescued in SB-3CT-treated and DENV-infected mice (Fig 6A–6C). Similarly, immunohistochemistry analyses also showed that β-catenin was attenuated by DENV infection compared with mock-infected mice in the liver (Fig 6D), spleen (Fig 6E), and lung (Fig 6F), but rescued in SB-3CT-treated and DENV-infected mice (Fig 6D–6F).

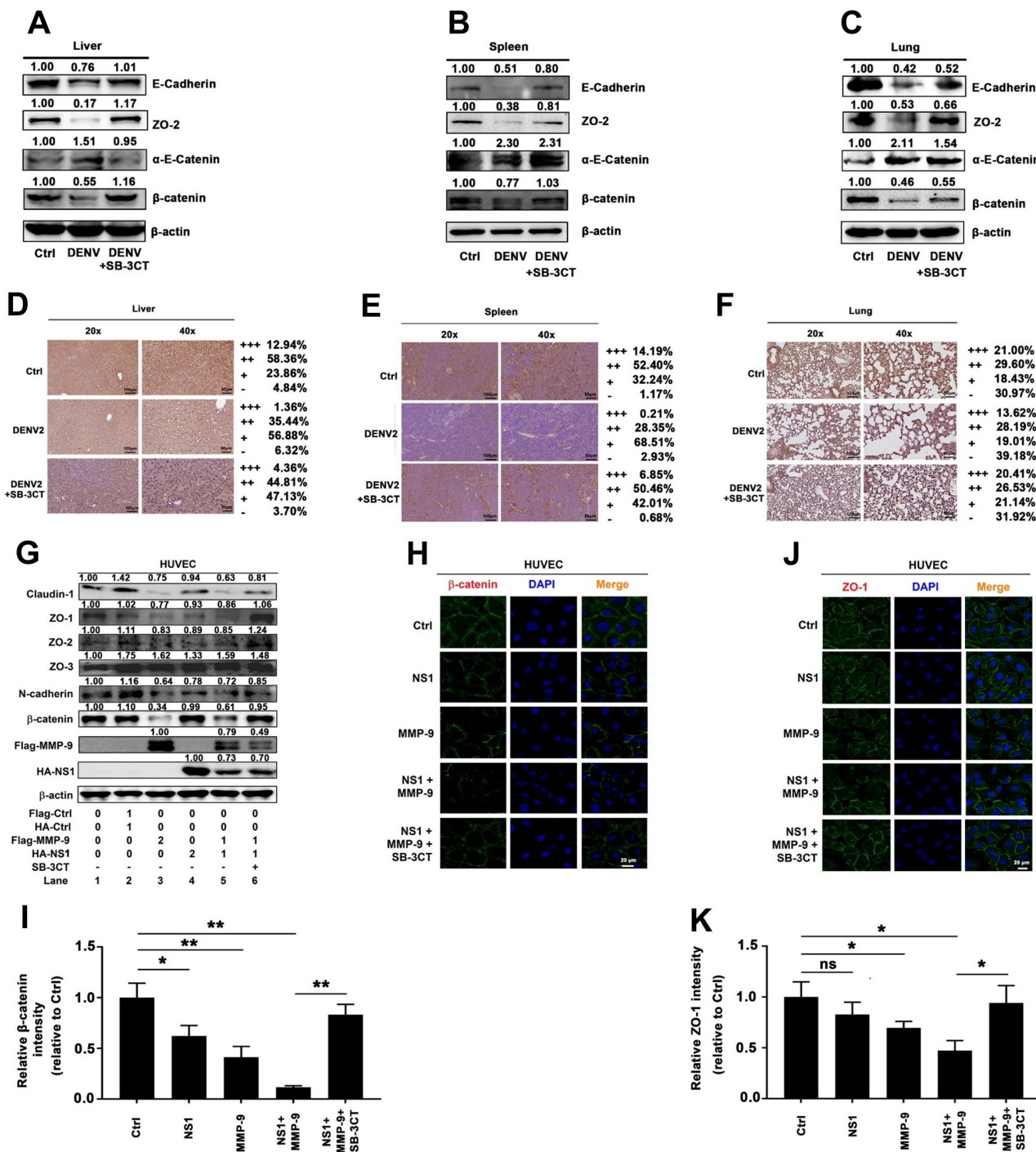

**Fig 6. NS1 recruits MMP-9 to disrupts the junctions between endothelial cells.** (**A–F**) IFNAR[-/-] C57BL/6 mice were intravenously injected with 300 μl DENV2 at a dose of 1×10[6] PFU/mouse (n = 6), pre-treated with 300 μl PBS containing MMP-9 specific inhibitor SB-3CT (5 mg/kg per mice) by intraperitoneal injection for 90 min and then treated with DENV2 (1×10[6] PFU/mouse), repeat treated with SB-3CT (5 mg/kg per mice) on the fourth day after DENV2 (NGC) infection (n = 6), or 300 μl PBS containing the same volume DMSO as a control group (n = 4). 7 days after infection, mice were euthanasia, and the tissues were collected. The indicated proteins in Lung (A), spleen (B) and Liver (C) were measured by Western-blot. The expression of β-catenin in Liver (D), spleen (E), and Lung (F) by Immunohistochemistry analysis. (**G**) HUVEC cells were respectively transfected with plasmid encoding MMP-9 (2 μg) or NS1 (2 μg) or NS1 (1 ug) plus MMP-9 (1 μg) for 24 h or firstly co-transfected with plasmid encoding NS1 (1ug) plus MMP-9 (1 μg) for 12 h, then treated with 600nM SB-3CT for 12 h. The indicated proteins in cell extract were analyzed by WB. (**H–K**) HUVEC cells were treated with NS1 protein (5 μg/ml) or MMP-9 protein (100 ng/ml) or NS1 (5 μg/ml) plus MMP-9 (100 ng/ml) or pre-incubated with 600 nM SB-3CT for 1 h, then treated with NS1 (5 μg/ml) plus MMP-9 (100 ng/ml) for 6 h, The distribution of endogenous β-catenin (H) or ZO-1 (J) protein were visualized

under confocal microscope. The quantifications of relative β-catenin (I) or ZO-1 (K) intensities were determined by ImageJ software. The quantification of protein was used by ImageJ software (A–G). (D–F) +++ means Percentage contribution of high positive cells; ++ means Percentage contribution of positive cells; + means Percentage contribution of low positive cells;—means Percentage contribution of negative cells. ns means not significant. All dates were representative of two to three independent experiments.

Taken together, our results showed that DENV2 induces vascular leakage through promoting MMP-9 to alter the adhesion and tight junctions in mice tissues.

Next, the effects of NS1 and MMP-9 on the regulation of adhesion and tight junction proteins were determined. The adhesion junction proteins such as N-cadherin and β-catenin, and the tight junction factors including Claudin-1 and Zona Occludens proteins (ZO-1, ZO-2, and ZO-3, also known as tight junction proteins, TJP-1, TJP-2, and TJP-3) were barely affected by NS1 protein alone in Hela cells (S5A Fig) or HUVECs (Fig 6G). In contrast, the tight junction protein ZO-1, ZO-2 and the adhesion junction protein β-catenin, N-cadherin were reduced by MMP-9 (2 μg) or NS1 (1 μg) plus MMP-9 (1 μg), and such reductions were eliminated by SB-3CT in both Hela (S5A Fig) and HUVEC cells (Fig 6G). Finally, the levels of endogenous β-catenin and ZO-1 proteins were visually evaluated by Immunofluorescence assays. Hela cells and HUVECs were incubated with commercialized DENV2 NS1 protein or recombinant human MMP-9 protein or NS1 protein plus MMP-9 protein. The levels of endogenous β-catenin (Figs S5B, S5C, 6H and 6I) and endogenous ZO-1 (Figs S5D, S5E, 6J and 6K) were significantly reduced by NS1 protein plus MMP-9 protein, and such reductions were recovered by SB-3CT in Hela cells (S5B–S5E Fig) and HUVECs (Fig 6H–6K).

## NS1 recruits MMP-9 to interact with adhesion and tight junction proteins

The mechanism by which NS1 and MMP-9 induce endothelial hyperpermeability and vascular leakage was evaluated. The interactions of NS1 and MMP-9 with adhesion and tight junction proteins were determined. Co-IP results showed that in HEK293T cells and Hela cells, NS1 and β-catenin interacted with each other (Figs 7A, 7B, S6A and S6B), and similarly, NS1 and ZO-1 associated with each other (Figs 7C, 7D, S6C and S6D). We noted that in HEK293T cells or Hela cells, MMP-9 and β-catenin failed to interact with each other (S6E–S6H Fig) and MMP-9 and ZO-1 were also unable to interact with each other (S6I and S6J Fig). His-pull-down assays further demonstrated that purified DENV2-NS1 protein could directly interact with β-catenin and ZO-1, but purified SARS-CoV-2 N protein and MMP-9 protein failed to interact with β-catenin and ZO-1 (Fig 7E and 7F). These results indicate that NS1 protein, but not N protein, can directly interact with β-catenin and ZO-1. Interestingly, in Hela cells in the presence of NS1, the MMP-9 protein and β-catenin protein could interact with each other (Fig 7G and 7H), and similarly MMP-9 and ZO-1 could also interact with each other (Fig 7I and 7J), indicating that NS1 facilitates the interaction of MMP-9 with β-catenin or ZO-1.

Moreover, the binding sites of NS1 involved in the NS1/MMP-9, NS1/β-catenin, NS1/ZO-1 interaction were determined by progressive truncation of NS1 (NS1-1-NS1-8) (S7A Fig). HEK293T cells were co-transfected with pHA-MMP-9 along with each of the plasmids expressing truncated NS1 protein, respectively. Co-IP results showed MMP-9 interacted with all the domains of NS1 (S7B Fig). HEK293T cells were transfected with each of the plasmids expressing truncated NS1 protein, respectively. Co-IP results showed that endogenous β-catenin interacted with NS1-1 (1–292), NS1-5 (72–352), NS1-6 (142–352), NS1-7 (212–52), and NS1-8 (282–352), but not with NS1-2 (1–212), NS1-3 (1–142), or NS1-4 (1–72) (S7C Fig), suggesting that the domain NS1-7 (212–352) are involved in NS1/β-catenin interaction. The results also indicated that endogenous ZO-1 interacted with NS1-5 (72–352), NS1-6 (142–352), NS1-7 (212–352), and NS1-8 (282–352), but not with NS1-1 (1–292), NS1-2 (1–212),

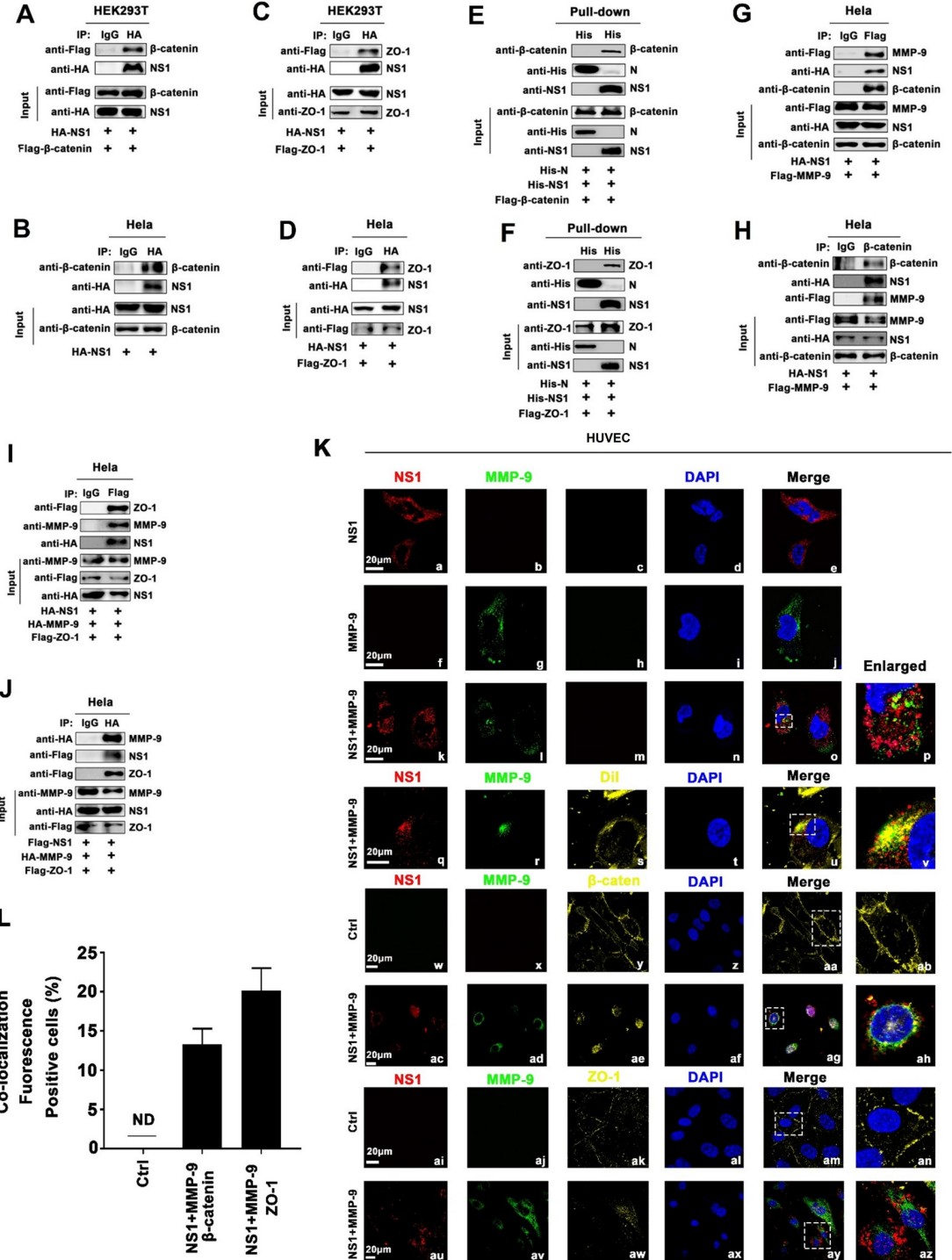

**Fig 7. NS1 recruits MMP-9 to interact with adhesion and tight junction proteins.** (**A**) HEK293T cells were transfected with plasmid encoding HA-NS1 plus Flag-β-catenin. Cell lysates were immunoprecipitated using anti-HA antibody, and analyzed using anti-Flag, anti-HA, or anti-β-catenin antibody. Cell lysates (40 μg) were used as Inputs. (**B**) Hela cells were transfected with plasmid encoding HA-NS1, Cell lysates were immunoprecipitated using anti-HA antibody, and analyzed using anti-HA or anti-β-catenin antibody. Cell lysates (40 μg) was used as Input. (**C**, **D**) HEK293T cells (C) or Hela cells (D) were co-transfected with plasmid encoding HA-NS1 plus Flag-ZO-1, Cell lysates were immunoprecipitated using anti-HA antibody, and analyzed using anti-Flag or anti-HA antibody. Cell lysates (40 μg) were used as Inputs. (**E**, **F**) HEK293T cells were transfected with Flag-β-catenin (E) or Flag-ZO-1 (F) and incubated with purified His-SARS-CoV-2-N (5 μg) or His-DENV2-NS1 (5 μg) for 24 h, cell extracts were incubated with Ni-NTA Agarose beads. Mixtures were analyzed by immunoblotting using anti-β-catenin, anti-NS1, anti-

His, anti-ZO-1 antibody. Untreated proteins including His-SARS-CoV-2-N (1 μg) and His-DENV2-NS1 (1 μg) and HEK293T cell lysates were analyzed by immunoblotting using anti-β-catenin, anti-NS1, anti-His, and anti-ZO-1 antibody (as input). (**G, H**) Hela cells were co-transfected with plasmid encoding HA-NS1 plus Flag-MMP-9, Cell lysates were immunoprecipitated using anti-Flag (G) or anti-β-catenin antibody (H), and analyzed using anti-Flag, anti-HA or anti-β-catenin antibody. Cell lysates (40 μg) was used as Input. (**I, J**) Hela cells were co-transfected with plasmid encoding HA-NS1, HA-MMP-9, and Flag-ZO-1 (I) or Flag-NS1, HA-MMP-9, and Flag-ZO-1 (J). Cell lysates were immunoprecipitated using anti-Flag (top) or anti-HA antibody (bottom), and analyzed using anti-Flag, anti-HA or anti-MMP-9 antibody. Cell lysates (40 μg) was used as Input. (**K, L**) HUVEC cells were treated with NS1 protein (5 μg/ml), MMP-9 protein (100 ng/ml), and NS1 protein (5 μg/ml) plus MMP-9 protein (100 ng/ml), respectively, for 6 h. The distributions of the membrane marker (Dil) (yellow), the endogenous β-catenin (yellow) and ZO-1 (yellow) proteins and the extracellular NS1 (red) and MMP-9 (green) proteins were visualized under confocal microscope. The quantifications of co-localization fluorescence were determined by ImageJ software (L). ND means not detected. All dates were representative of three independent experiments.

NS1-3 (1–142), or NS1-4 (1–72) (S7C Fig), suggesting that the domain NS1-8 (282–352) are involved in NS1/ZO-1 interaction.

We next constructed three new truncated mutants in accordance with the structural domain of NS1 (S7D Fig). Similar to the previous results, MMP9 could interact with different structural domains of NS1 (Domain 1 of NS1 was not detected here because amino acids were too small to be precipitated with MMP9) (S7E Fig), but β-catenin/ZO-1 only interacts with the C-terminal β-ladder domain of NS1 (S7F Fig). Finally, HUVECs were incubated with commercialized DENV2 NS1 protein, recombinant human MMP-9 protein, and NS1 protein plus MMP-9 protein, respectively. Immunofluorescence results revealed that when presented individually, NS1 protein (Fig 7K a-e) or MMP-9 protein (Fig 7K f-j) was distributed outside the nucleus; when precented together, extracellular NS1 protein and MMP-9 protein were co-localized with each other (Fig 7K k-p) and distributed in cell membrane (Fig 7K q-v). Notably, NS1 and MMP-9 protein also co-located with β-catenin (Fig 7K ac-ah) and ZO-1 (Fig 7K au-az). We also noticed that compared with untreated cells (Fig 7K w-ab and 7K ai-an), the structures of β-catenin (Fig 7K ac-ah) and ZO-1 (Fig 7K au-ay) were destroyed in the presence of NS1 and MMP-9 in HUVEC cells (Fig 7K and 7L). There results confirm that NS1 recruits MMP-9 to disrupt the junctions between endothelial cells. Therefore, our results reveal that NS1 is probably acting as bridge to promote the interactions of MMP-9 with adhesion and tight junction proteins.

## NS1 induce vascular leakage through recruiting MMP-9 in mice

Finally, the effects of NS1 and MMP-9 on the induction of vascular leakage were determined in wild-type (C57BL/6) and MMP-9 deficient (MMP-9$^{-/-}$) mice. First, the knockout of the MMP-9 gene was confirmed by genotyping using mouse tail DNA samples (S8 Fig). Next, C57BL/6 mice (n = 6, six-week-old) and MMP-9$^{-/-}$ mice (n = 6, six-week-old) were injected (*i. v.*) with DENV2 NS1 protein and/or recombinant mouse MMP-9 protein, followed by intravenous injection with Evans blue dye. At 24 h post-treatment, the mice were euthanized, and mouse tissues were collected. Strikingly, the intensities of Evans blue dye were enhanced by NS1 protein in the lung (Fig 8A), spleen (Fig 8B), and liver (Fig 8C) of C57BL/6 mice. However, the intensities of Evans blue dye were relatively unchanged by NS1 in the tissues of MMP-9$^{-/-}$ mice (Fig 8A–8C). Interestingly, the intensities of Evans blue dye were significantly enhanced by NS1 in the tissues of MMP-9$^{-/-}$ mice treated with MMP-9 (Fig 8A–8C). Therefore, the results reveal that DENV NS1 induces vascular leakage in mouse tissues, and demonstrate that MMP-9 is required for NS1-induced vascular leakages in mice. Moreover, the effects of NS1 and MMP-9 on the production of adhesion and tight junction proteins in mice were also determined. C57BL/6 mice (n = 6, six-week-old) and MMP-9$^{-/-}$ mice (n = 6, six-week-old) were intravenously (via tail vein) injected with DENV2 NS1 protein and/or

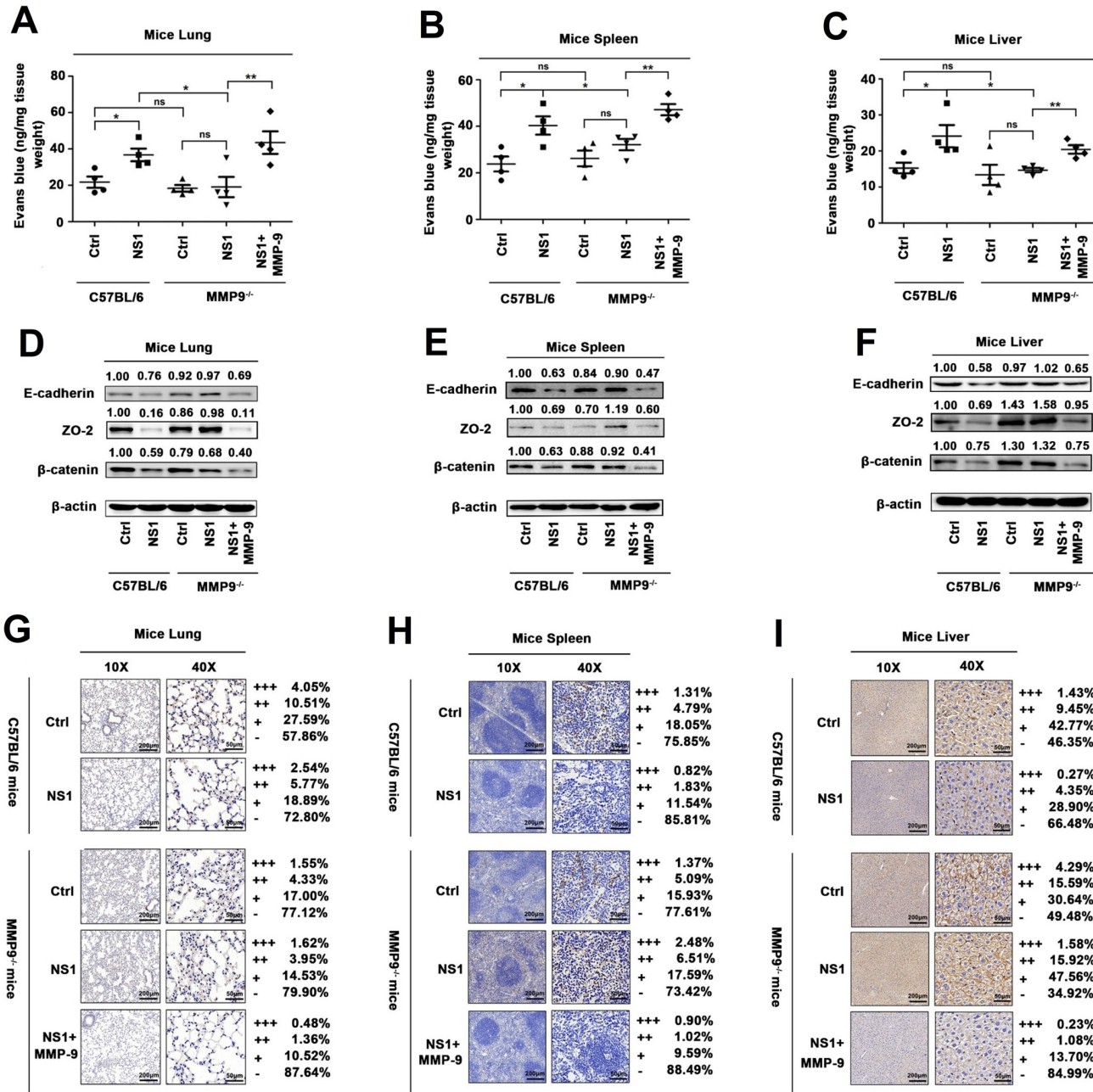

**Fig 8. NS1 induce vascular leakage through recruiting MMP-9 in mice.** C57BL/6 mice and MMP-9[-/-] mice were injected intravenously DENV2 NS1 protein [10 mg/kg (n = 5)], the same volume of PBS was also tail vein injected to C57BL/6 mice and MMP-9[-/-] mice (n = 5) as control group. Another group of MMP-9[-/-] mice (n = 5) were injected intravenously DENV2 NS1 protein (10 mg/kg) plus recombinant mouse MMP-9 protein (70 µg /kg). (**A–C**) After 24 h post-injection, mice were intravenously injected with Evans blue dye. The dye was allowed to circulate for 2h before mice were euthanized, and tissue include Lung (A), spleen (B), and Liver (C) were collected. The value of Evans blue was measured at $OD_{610}$. (**D–I**) After 24 h post-injection, mice were euthanized and tissue were collected. The indicated proteins in Lung (D), spleen (E), and Liver (F) were measured by Western-blot. The expression of β-catenin in Lung (G), spleen (H), and Liver (I) were analyzed by Immunohistochemistry. All dates were representative of two to three independent experiments. The quantification of protein was used by ImageJ software (D–I). (G–H) +++ means Percentage contribution of high positive cells; ++ means Percentage contribution of positive cells; + means Percentage contribution of low positive cells;—means Percentage contribution of negative cells. ns means not significant. Values are mean ± SEM, P ≤0.05 (*), P ≤0.01 (**), P ≤0.001 (***).

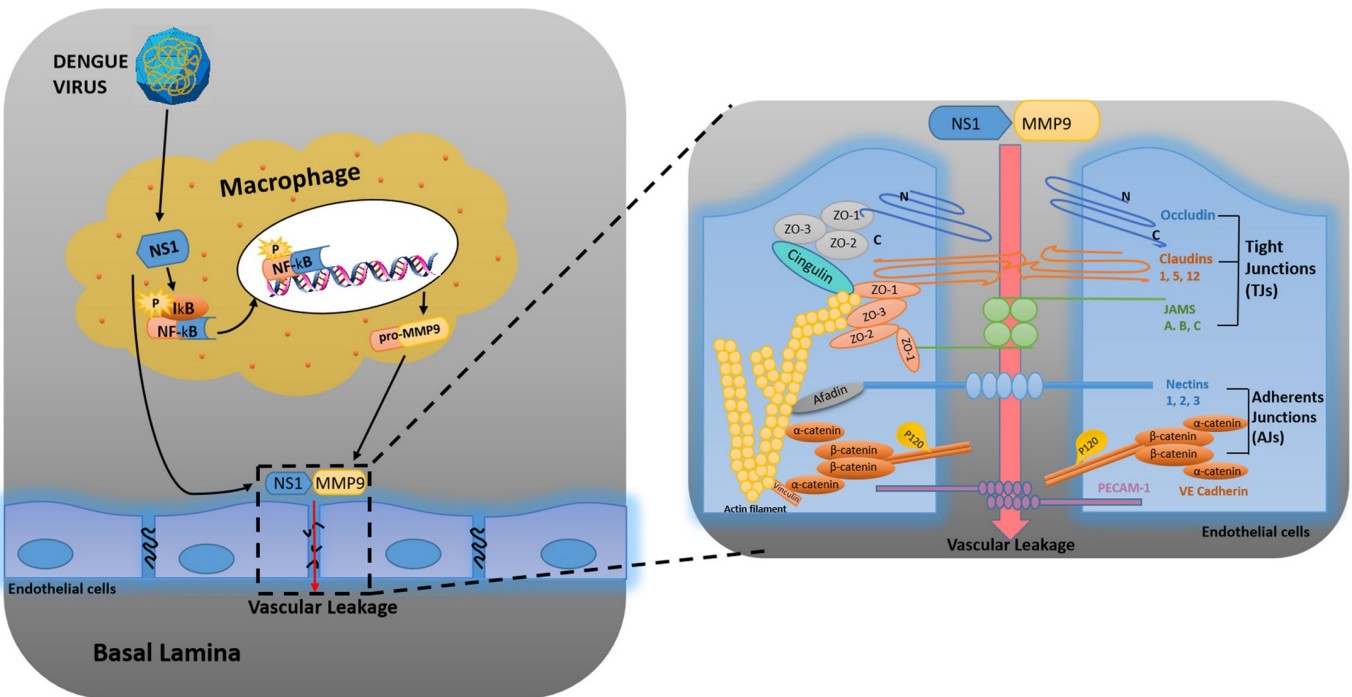

**Fig 9. A proposed model in which DENV NS1 and MMP-9 coordinate to induce vascular leakage by altering endothelial cell adhesion and tight junctions.** DENV non-structural protein 1 (NS1) induces MMP-9 expression through activating the nuclear factor κB (NF-κB) signaling pathway. Additionally, NS1 interacts with MMP-9 and facilitates the enzyme to alter the adhesion and tight junctions and vascular leakage in human endothelial cells and mice tissues. Moreover, NS1 recruits MMP-9 to interact with β-catenin and Zona occludens protein-1/2 to degrade the important adhesion and tight junction proteins, thereby inducing endothelial hyperpermeability and vascular leakage in human endothelial cells and mice tissues.

recombinant mouse MMP-9 protein. At 24 h post-treatment, the mice were euthanized, and mouse tissues were collected. β-catenin and ZO-2 proteins were down-regulated by NS1 in the lung (Fig 8D), spleen (Fig 8E), and liver (Fig 8F) of C57BL/6 mice, relatively unaffected by NS1 in MMP9$^{-/-}$ mouse tissues, and significantly reduced by NS1 in the tissues of MMP9$^{-/-}$ mice treated with MMP-9 protein (Fig 8D–8F). Similarly, immunohistochemistry staining showed that β-catenin reduced by NS1 in the lung (Fig 8G), spleen (Fig 8H), and lives (Fig 8I) of C57BL/6 mice, relatively unaffected by NS1 in the tissues of MMP9$^{-/-}$ mice, and significantly reduced by NS1 in the tissues of MMP9$^{-/-}$ mice treated with MMP-9 protein (Fig 8G–8I). Taken together, our findings demonstrate that NS1 induced endothelial hyperpermeability in HUVECs and vascular leakages in mice, and revealed that MMP-9 is required for NS1-induced endothelial hyperpermeability and vascular leakage (Fig 9).

## Discussion

DENV infection may cause life-threatening diseases such as DHF, DSS, and ADE [30, 31]. The clinical symptoms of DENV infection include hypotension, reduced blood volume, and vascular permeability changes [32]. Therefore, it is important to investigate the mechanism by which DENV infection increases vascular permeability. The present study revealed a distinct mechanism by which DENV induces endothelial permeability and vascular leakage in human endothelial cells and mice tissues.

Our initial results show that DENV2 promotes MMP-9 expression and secretion in human PBMCs and macrophages. These results are consistent with previous reports that MMP-9 protein is highly expressed in immune cells [15], but expressed at a very low level in endothelial

cells [22]. More significantly, our findings demonstrate that DENV NS1 enhanced MMP-9 expression through the activation of the NF-κB signaling pathway and modulate the proteolytic activity through interacting with MMP-9. Other groups also proved that ZIKV NS1 protein bound to MMP-9 and facilitated K63-linked polyubiquitination of MMP-9 to stabilize the expression of MMP9, leading to the destruction of the blood-testis barrier [33]. Previous studies revealed that the surface of the endothelial cells (ECs) is coated with a glycocalyx of membrane-bound macromolecules comprised of sulfated proteoglycans, glycoproteins, and plasma proteins that adhere to the surface matrix [12], MMP-9 is destructive to the integrity of endothelial cells and specifically degrades extracellular matrix [14, 16], and DENV induces vascular leakage by up-regulating the expression of MMP-9 in Dendritic cells (DCs) [17]. Here, we further demonstrate that NS1 interacts with the Zinc-binding catalytic domain and the Fibronectin type-like domain of MMP-9 and facilitate the enzyme to alter the adhesion and tight junctions, and thereby promoting vascular leakage, in human endothelial cells and mice tissues including liver, spleen, and lung.

The maintenance of endothelial cell permeability is determined by two factors: (1) EGLs form protective membranes on the surfaces of endothelial cells [11, 12], and (2) the adhesion and tight junctions between endothelial cells maintain the integrity of endothelial cells [13]. It was reported that NS1 induces vascular endothelial cell permeability leading to vascular leakage by disrupting the extracellular polysaccharide-protein complexes [19, 20, 24], and NS1 acts as a pathogen-associated molecular pattern (PAMP) by activating the Toll-like receptor 4 (TLR4) signaling pathways to promote the release of inflammatory factors IL-6 and IL-8 and induce vascular leakage [20]. Recent studies have further confirmed that macrophage migration inhibitory factor (MIF) plays a key role in regulating NS1-induced degradation of extracellular polysaccharide proteins [22].

Interestingly, here we demonstrate that NS1 induces endothelial hyperpermeability in human endothelial cells and mice tissues through activating MMP-9. The levels of TEER are reduced by MMP-9 and NS1, but the reductions are recovered by the specific inhibitor of MMP-9 (SB-3CT), demonstrating that MMP-9 induces endothelial hyperpermeability in human endothelial cells. Additionally, the intensities of Evans blue dye are enhanced by NS1 in the tissues of C57BL/6 mice; relatively unaffected by NS1 in the tissues of MMP-9$^{-/-}$ mice; and significantly facilitated by NS1 in the tissues of MMP-9$^{-/-}$ mice treated with MMP-9 protein; revealing that NS1 induces vascular leakage in mice tissues, and demonstrating that MMP-9 is required for NS1-induced hyperpermeability in mice tissues.

More interestingly, the mechanism by which NS1 and MMP-9 induce endothelial hyperpermeability and vascular leakage is further revealed. The adhesion junction proteins N-cadherin and β-catenin as well as the tight junction factors ZO-1, ZO-2, and ZO-3 are not affected by NS1, but ZO-1, ZO-2, N-cadherin, and β-catenin are reduced by MMP-9, and such reductions are eliminated by SB-3CT. Similarly, the endogenous ZO-1, ZO-2, N-cadherin, and β-catenin are reduced by NS1 and MMP-9, and such reductions are recovered by SB-3CT. Additionally, β-catenin and ZO-2 are down-regulated by NS1 in of WT C57BL/6 mice tissues, relatively unaffected by NS1 in MMP9$^{-/-}$ mice tissues, while significantly repressed by NS1 in the tissues of MMP9$^{-/-}$ mice supplemented with MMP-9 protein. Moreover, we further reveal that NS1 can interact with β-catenin and ZO-1 through different domains; MMP-9 fails to interact with β-catenin and ZO-1; however, in the presence of NS1, MMP-9 associates with β-catenin and ZO-1; indicating that NS1 facilitates MMP-9 to interacting with β-catenin or ZO-1; and thereby degrading the adhesion and tight junction proteins. However, the detailed functional relevance of the interaction between NS1 and MMP-9 needs further investigations in the future. Taken together, these results demonstrate that NS1 induces hyperpermeability and vascular leakages in endothelial cells and mice tissues, and reveal that MMP-9 is required for

NS1-induced endothelial hyperpermeability and vascular leakage through degrade ting the adhesion and tight junction proteins. In our previous work, we found that DENV M protein induced vascular leakage in mice by activating NLRP3 inflammasome [9, 34], these two different molecular mechanisms are induced in DENV infection maybe play an equally important role in vascular leakage.

In summary, we reveal a distinct molecular mechanism by which the viral NS1 protein coordinates with the host factor MMP-9 to induce endothelial hyperpermeability and vascular leakage in human endothelial cells and mice tissues through disrupting the adhesion and tight junctions between endothelial cells. This study would provide new in signs into the pathogenesis caused by DENV infection, and suggest that MMP-9 may acts as a drug target for the prevention and treatment of DENV-associated diseases.

## Materials and methods

### Ethics statement

All human subjects were adult. The study was conducted according to the principles of the Declaration of Helsinki and approved by the Institutional Review Board of the College of Life Sciences, Wuhan University in accordance with its guidelines for the protection of human subjects. The Institutional Review Board of the College of Life Sciences, Wuhan University, approved the collection of blood samples for this study, and it was conducted in accordance with the guidelines for the protection of human subjects. Written informed consent was obtained from each participant.

All animal studies were performed in accordance with the principles described by the Animal Welfare Act and the National Institutes of Health Guidelines for the care and use of laboratory animals in biomedical research. All procedures involving mice and experimental protocols were approved by the Institutional Animal Care and Use Committee (IACUC) of the College of Life Sciences, Wuhan University (Permit numbers: WDSKY0201901).

### Clinical sample analysis

In this study, severe dengue patient samples were collected at Eight people's Hospital of Guangzhou during a DENV outbreak in Guangzhou, China, in 2014. Patients were categorized as having severe dengue according to the 2009 WHO criteria for dengue severity. The characteristics of sever dengue patients were listed in S1 Table. All the dengue patient samples were assessed by anti-dengue IgM and IgG enzyme-linked immunosorbent assay (ELSIA) and qRT-PCR to DENV nucleic acid test. Serum samples from 8 patients with severe dengue were collected for ELISA analysis on day 2, day 5, day 8, and day 11 after hospitalization. The expression of NS1 and MMP-9 in individual dengue patient were listed in S2 Table. In addition, 8 serum samples from healthy donors were include as the negative control. Informed consent was obtained from each person.

### Animal studies

Wide-type (WT) C57BL/6 mice were purchased from Hubei Research Center of Laboratory Animals (Wuhan, Hubei, China). MMP9$^{-/-}$ mice were purchased from Model Animal Research Center of Nanjing University. *IFNAR*$^{-/-}$—C57BL/6 mice were bred in our laboratory. All mice were bred and maintained under specific pathogen-free conditions at Jinan University. For DENV2 infection assays, 6-week-old *IFNAR*$^{-/-}$ C57BL/6 mice were tail vein injected with PBS (mock infection), pre-treated with 300 μl PBS containing MMP-9 specific inhibitor SB-3CT (5 mg/kg per mice) by intraperitoneal injection for 90 min and then treated with

DENV2 ($1\times10^6$ PFU/mouse), repeat treated with SB-3CT (5 mg/kg per mice) on the fourth day after DENV2 (NGC) infection, or 300 μl PBS containing the same volume DMSO as a control group. One week after the DENV2 injection, mice were sacrificed, and tissues were collected for immunohistochemical and histopathological analyses. For other animal assays, six-week-old and sex-matched of MMP9$^{-/-}$ mice and Wide-type C57 BL/6 mice were randomly chosen to injection with DENV2-NS1 protein or recombinant mouse MMP-9 protein or DENV2-NS1 protein plus MMP-9 protein. The equivalent volumes of PBS-injected mice were used as negative controls. At 24 h post-injection, mice were sacrificed, and tissue were collected for Histopathology analysis.

## Cell culture and transfection

Human umbilical vein endothelial cells (HUCEV) were purchased form Obio Technology (Shanghai, China). Human monocytic cell lines (THP-1), human embryonic kidney cell lines (HEK-293T), Hela cells, African green monkey cell lines (Vero) and Aedes albopictus gut cell lines (C6/36) were purchased from the American Type culture Collection (ATCC). THP-1 was grown in RPMI 1640 medium supplemented with 10% fetal calf serum, 100 U/ml penicillin, and 100 μg/ml streptomycin sulphate. HEK-293T, HUVEC and Vero were grown in DMEM medium with 10% fetal calf serum, 100 U/ml penicillin, and 100 μg/ml streptomycin sulphate. C6/36 was grown in MEM medium supplemented with 10% fetal calf serum, 100 U/ml penicillin, and 100 μg/ml streptomycin sulphate. THP-1, HEK-293T, HUVEC and Vero cells were maintained at 37˚C in a 5% $CO_2$ incubator. C6/36 were maintained at 30˚C in a 5% $CO_2$ incubator. In order to differentiation of macrophages. THP-1 were stimulated with Phorbol-12-myristate-13-acetate (PMA) for 12 h. Afterwards, cells were incubated for 24 h without PMA. Among those cells, HEK293T and Hela cells are easy to be transfected with plasmids, and the main purpose of whose are to verify the protein expression and the interaction between proteins. THP-1 and HUVEC cells are mainly used to study DENV virus infection and changes in endothelial cell permeability. Vero and C6/36 cells are used to amplification of DENV virus.

Peripheral blood mononuclear cells (PBMCs) were separated by density centrifugation of fresh peripheral venous blood samples that they were diluted 1:1 in pyrogen-free PBS over Histopaque (Haoyang Biotech). Then the cells were washed twice with PBS and resuspended in medium (RPMI 1640) supplemented with 10% FBS, penicillin (100 U/ml), streptomycin (100 μg/ml) in 6 well plates and 12 well plates.

Lipo2000 (Invitrogen) were used for cell transfection in HEK293T cells, Hela cells, THP-1 cells, and HUVECs. Take HUVECs transfection as an example. The operation was as follows: when HUVEC cells grew to 80–90% of the volume of the cell culture dish, the media were replaced with serum-free medium, and transfected with the ratio of plasmid concentration (1 μg) and Lipo2000 dosage 1:3. After transfection, HUVEC cells were replaced with complete medium until the predetermined time point.

## Virus

All experiments used DENV-2 strain NGC (GenBank accession number KM204118.1) was kindly provided by Dr. Xulin Chen of Wuhan Institute of Virology, Chinese Academy of Sciences. To generate large stocks of dengue virus for experiments. C6/36 cells or Vero cells were incubated with DENV-2 at MOI of 0.5 for 2 h, then unbound dengue virus was washed away. The infected cells were cultured sequentially in fresh medium with 2% FBS until seven days. Supernatants were harvested and centrifuged at 4000 rpm for 10 min to remove cellular debris; then they were filtrated by 0.22 μm filter membrane. All dengue virus was aliquoted into tubes for freezing at -70˚C. Virus tiles were determined by plaque assay using Vero cells.

## Regents and antibodies

Phorbol-12-myristate-13-acetate (PMA), gelatin, Triton X-100, Coomassie brilliant blue R-250 was purchased from Sigma-Aldrich, MMP-9 inhibitor (SB-3CT) and NF-κB inhibitor (sc-75741) were purchased from Selleck. Recombinant human MMP-9 protein and Recombinant mice MMP-9 protein were purchased from R&D systems. Recombinant human pro-MMP-9 protein were purchased from ProSpec. Commercialized DENV2-NS1 protein were purchased from Native Antigen. Trizol reagent was purchased from Ambion. Lipofectamine 2000 reagent was purchased from Invitrogen. Human MMP-9 ELISA kit was purchased from BD Biosciences. Membrane maker Dil (D8700) was purchased from Solarbio. NF-κB Pathway Sampler Kit (#9936T), Tight Junction Antibody Sampler Kit (#8683T), Cadherin-Catenin Antibody Sampler Kit (#9961T), and Antibodies against TIMP-1 (8946S) were purchased from Cell Signaling Technology. Antibody against DENV-NS3 (GTX124252) were purchased from Genetex. Antibodies against DENV2-NS1 were purchased from Arigo biolaboratories (SQab1501). Antibodies against Flag (F3165) and HA (H6908) were purchased from Sigma. Anti-β-actin antibody (66009) were purchased from Proteintech. Rabbit IgG and Mouse IgG were purchased from Invitrogen. Anti-Mouse IgG Dylight 649, Anti-Mouse IgG Dylight 488, Anti-Rabbit IgG Dylight 649, and Anti-Rabbit IgG FITC were purchased from Abbkine.

## RNA extract and Real-time PCR

Trizol reagent (Invitrogen, Carlsbad, CA) was used for total cellular RNA extracted according to the manufacturer's instructions. The RNAs (1 μg) were then reverse transcribed to cDNA with 0.5 μl oligo (dT) and 0.5 μl Random primer at 37°C for 60 min and 72°C for 10 min. The cDNA then was used as templates for real-time PCR analysis. Real-time PCR was performed in a LightCycler 480 thermal cycler (Roche) by the following procedure: heat activate polymerase at 95°C for 5 min, afterwards, 45 cycles of 95°C for 15s, 58°C for 15s and 72°C for the 30s, the fluorescence was collected and analyzed at the 72°C step. A final melting curve step from 50°C to 95°C was used to test the specificity of the primer. The primers used in real-time PCR detection were listed S3 Table

## Plasmids' construction

Plasmids pcDNA3.1(+)-3×flag-Cap/prm/M/E/NS1/NS2A/NS2B/NS3/NS4A/NS4B/NS5/ MMP-9 and plasmid pCAggs-HA-MMP-9/NS1 were constructed previously by our laboratory. The coding regions of β-catenin and ZO-1 were generated by PCR amplification. For β-catenin, the PCR production was inserted into the *Bam*HI and *Xba*I sites of Plasmid pcDNA3.1(+)-3×flag. For ZO-1, the PCR production was inserted into the *Kpn*I and *Eco*RV sites of Plasmid pcDNA3.1(+)-3×flag. For the truncated forms of MMP-9, the PCR productions were inserted into the *Eco*RI and *Xho*I sites of Plasmid pcDNA3.1(+)-3×flag. The sequences of primers were shown in S4 Table.

## Zymography assay

MMP-9 proteinase activity was detected by the gelatin zymography assay as described previously. Briefly, the cells supernatants were separated in SDS-PAGE gels containing 1mg/ml gelatin. Then the gel was washed 3 times with 2.5% Triton X-100 (45 min every time), followed the gel was washed with 50 mM Tris-HCl (pH7.6) containing 5 mM CaCl$_2$, 1 μM ZnCl$_2$ and 0.02% Sodium Azide for 30 min. Afterwards, the gel was incubated overnight at 37°C in the same buffer, then the gel was stained with 0.25% Coomassie brilliant blue R-250 for 2 h and then distaining.

## Enzyme-linked immunosorbent assay

The concentration of culture supernatants and MMP-9 or NS1 were measured by Human MMP-9 ELISA Kit (Invitrogen) or DENV2-NS1 ELISA Kit (Arigo biolaboratories) according to manufacturer's instructions.

## Biomolecular fluorescence complementation (BiFC) assay

The N-terminal truncated (VN-173) and the C-terminal truncated (VC-155) version of non-fluorescent Venus YFP fragments vectors were brought here (Addgene plasmid # 22010). Human MMP-9 gene was subcloned in VC-155, and DENV NS1, E, and NS4A genes were subcloned in VC-155 respectively. The resulting plasmids or empty vectors were co-transfected into HEK293T cells or Hela cells using Lipo2000. At 24 h post-transfection, cells were pre-culture at 4˚C for 10 min. The fusion proteins in living cells were observed by confocal microscopy.

## Yeast double hybridization assay

Human MMP-9 gene was subcloned in pGBKT7, and DENV NS1, E, and NS4A were subcloned in pGADT7 respectively. control vectors pGBKT7, pGADT7, pGBKT7-p53, pGADT7-T, and pGBKT7-lam, and some reagents were purchased from Clontech Laboratories. All experimental procedures were done following the Matchmaker Gold Yeast 2-Hybrid System User Manual. Briefly, yeast strain AH109 cells were co-transformed with plasmid pGADT7 and plasmid pGBKT7. Transformed yeast cells were grown in SD-minus Trp/Leu plates (DDO) for 24–48 h, and then subcloned replica plated on SD-minus Trp/Leu/Ade/His plate (QDO) for another 48–96 h.

## Western-blot

The PMA-differentiated THP-1 cells were collected and then washed twice with PBS and dissolved in THP-1 lyses buffer (50 mM Tris-HCl, 150 mM NaCl, 0.1% Nonidetp40, 5 mM EDTA, and 10% glycerol, pH7.4). The HEK-293T cells, HUVEC cells and Hela cells were prepared in 293T lyses buffer (50 mM Tris-HCl, 300 mM NaCl, 1% Triton X-100, 5 mM EDTA, and 10% glycerol, pH7.4). 10% protease inhibitor (Roche) were added to lyses buffer before using. Protein concentration was measured by the Bradford assay (Bio-Rad, Richmond, CA). Cultured cell lysates (50 μg) were electrophoresed in an 8–12% SDS polyacrylamide gels and transferred to nitrocellulose membranes (Amersham, Piscataway, NJ). Nonspecific bands of NC membranes were blocked by using 5% skim milk for 2 h. Then membranes were washed third with phosphate buffered saline with 0.1% Tween 20 (PBST) and incubated with the specific antibody. Protein bands were visualized using a Luminescent Image Analyzer (Fujifilm LAS-4000).

## Co-immunoprecipitation assays

HEK-293T cells or Hela cells were spread to 6-cm-diameter dishes, and co-transfected with the purpose of plasmids for 24 h. Then the cells were lysed in 293T lyses buffer (50 mM Tris-HCl, 300 mM NaCl, 1% Triton X-100, 5 mM EDTA, and 10% glycerol, pH7.4), The lyses buffer was rotating at 4˚C for 30 min and centrifuged at 12000 rpm for 15 min to remove cellular debris. A little part of supernatants was sucked out as Input, and the others were incubated with the indicated antibodies overnight at 4˚C.Then mixed with the Protein G sepharose beads (GE Healthcare) for 2 h at 4˚C.The immunoprecipitates were washed four to six times with the 293T lyses buffer (50 mM Tris-HCl, 300 mM NaCl, 1% Triton X-100, 5 mM EDTA,

and 10% glycerol, pH7.4), boiled in protein loading buffer for 10 min. Then analyzed by using SDS-PAGE and Western blotting.

## His-pull-down assays

Commercial purified DENV2-NS1 were purchased from Sino Biological (40243-V07H) and Commercial purified MMP-9 protein were purchased from RD company (P14780). To construct pET-28a-His-SARS-CoV-2-N, N gene was sub-cloned into pET-28a-His at EcoRI and XhoI sites. Plasmid pET-28a-His-N was transfected into Escherichia coli strain BL21. After growing in Kanamycin-resistant LB medium at 37˚C until $OD_{600}$ reached 0.6–0.8, IPTG was added to a concentration of 0.2 mM, and medium was transferred to 16˚C for 12–16 h. Cells were harvested and sonicated in lysis buffer (50 mM $NaH_2PO_4$, 100 mM PMSF, 300 mM NaCl, 10 mM imidazole, pH8.0). Lysates were centrifuged at 12000 rpm for 15 min to move debris. Supernatants were loaded into Ni-NTA Agarose columns (QIAGEN), shaken at 4˚C for 2 h, and slowly flowed from columns and washed twice with wash buffer (50 mM $NaH_2PO_4$, 100 mM PMSF, 300 mM NaCl, 20 mM imidazole, pH8.0). Recombinant His-N protein was eluted using elution buffer (50 mM $NaH_2PO_4$, 100 mM PMSF, 300 mM NaCl, 250 mM imidazole, pH8.0). Recombinant His-N containing reduced imidazole was replaced into PBS using Millipore ultrafiltration tube. Ni-NTA Agarose were incubated with purified DENV2-NS1 or His-N at 4˚C for 2 h, and incubated with purified MMP-9 protein or cell lysates from HEK293T cells transfected with pcDNA3.1(+)-3×flag-β-catenin/ZO-1 for 16 h at 4˚C. Precipitates were washed 4 times with PBS, boiled in protein loading buffer, and separated by SDS-PAGE.

## Immunofluorescence

HEK-293T cells were grown on sterile cover slips were transfected with HA-MMP-9 and Flag-NS1 at 40% confluence for 24 h. HUVEC cells or Hela cells were grown on sterile cover slips at 80% confluence, then treated with NS1 protein (5 μg/ml) or MMP-9 protein (100 ng/ml) or NS1 (5 μg/ml) plus MMP-9 (100 ng/ml) or pre-incubated with 600 nM SB-3CT for 1 h, then treated with NS1 (5 μg/ml) plus MMP-9 (100 ng/ml) for 6 h. Cells were fixed with 4% paraformaldehyde for 15 min and then washed three times with wash buffer (ice-cold PBS containing 0.1% BSA), permeabilized with PBS containing 0.2% TritonX-100 for 5 min and washed three times with wash buffer, after blocking with 5% BSA for 30 min, cells were incubated overnight with anti-HA antibody and anti-Flag antibody (1:200 in wash buffer) or anti-β-catenin or anti-ZO-1 antibody (1:100 in wash buffer), followed by staining with FITC-conjugated donkey anti-mouse IgG and Daylight 649-conjugated donkey anti-rabbit IgG or just Cy3-conjugated donkey anti-mouse IgG secondary antibody (Abbkine) (1:100 in wash buffer) for 1 h. Nuclei were stained with DAPI for 5 min, and then the cells were washed three times with wash buffer. Finally, the cells were viewed using a confocal fluorescence microscope (Fluo View FV1000; Olympus, Tokyo, Japan).

## Quantization of vascular leakage in vivo

The level of vascular leakage in mice was quantified through Evans blue assays as previously described. Briefly, 300 μl of 0.5% Evans blue dye was injected intravenously to five groups and allowed the dye to circulate for 2 h. Then the mice were euthanized and extensively perfused with PBS. Tissues was collected and weighed. The tubes containing tissue were added to 1 ml formamide and incubated at 37˚C for 24 h. Evans blue concentration was quantified by measuring $OD_{610}$ and comparing to the standard curve. Data was expressed as ng Evans blue dye/ mg tissue weight.

## Trans-endothelial electrical resistance (TEER)

Human umbilical vein endothelial cells (HUVEC) monolayers ($2 \times 10^5$) were grown on the 24-well Transwell polycarbonate membrane system (Transwell permeable support, 0.4 μm, 6.5 mm insert; Corning Inc.) until a monolayer formed (about 24 h), and then treated with different reagents. After 24 h of treatment, 50% of upper and lower chamber media was replaced by fresh endothelial cell medium. Untreated endothelial cells grown on Transwell inserts were used as negative controls and medium alone were used for blank controls. Endothelial permeability was evaluated by measuring TEER in ohms at indicated time points using EVOM2 epithelial voltohmmeter (World Precision Instruments). Relative TEER was expressed as follows: [ohm (experimental groups)—ohm (medium alone)] / [ohm (untreated endothelial cells)—ohm (medium alone).

## Statistical analysis

All experiments were repeated two to three times with similar results. All results were expressed as the mean ± the standard deviation (SD). Statistical analysis was carried out using the t-test for two groups and one-way ANOVA for multiple groups (GraphPad Prism5). The date was considered statistically significant when $P \leq 0.05$ (*), $P \leq 0.01$ (**), $P \leq 0.001$ (***).

## Supporting information

**S1 Fig. DENV NS1 interacts with MMP-9, but DENV E and NS4A not interacts with MMP-9.** (**A**, **B**) HEK293T cells and Hela cells were was co-transfected with empty vector or VC-155-MMP-9 and VN173-NS1/E/NS4A. At 24 h post-transfection, cell lysates were analyzed by immunoblotting. (**C**, **D**) Hela cells were was co-transfected with empty vector or VC-155-MMP-9 and VN173-NS1/E/NS4A. At 24 h post-transfection, living cells were observed by confocal microscopy (**C**). The quantification of YFP-positive cells was used by ImageJ software (**D**). ND means not detected. (**E**, **F**) Yeast strain AH109 were co-transformed with combination of binding domain (BD-MMP-9) and activation domain (AD-NS1, AD-E, and AD-NS4A) plasmid. Transfected yeast cells were grown on SD-minus Trp/Leu double dropout plates, and colonies were replicated on to SD-minus Trp/Leu/Ade/His fourth dropout plates to check for the expression of reporter genes (**E**). Yeast strain AH109 were co-transformed with combination of binding domain (BD-MMP-9) and activation domain (AD-NS1, AD-E, and AD-NS4A) plasmid. At 48 h post-transfection, cell lysates were analyzed by immunoblotting (**F**). Dates were representative of two to three independent experiments. (TIF)

**S2 Fig. The expression of NS1 was detected in cultured cells and DENV2 E and NS3 had no influence in MMP-9 expression.** (**A–H**) HEK293T cell (A, E), Hela cells (B, F), PMA-differentiated THP-1 macrophages (C, G) and HUVECs (D, H) were transfected with the different concentrations of plasmid encoding *NS1* for 24 h. NS1 protein in Supernatants were analyzed by ELISA (A–D). Cell lysates were analyzed (E–H) by immunoblotting. (**I**, **J**) PMA-differentiated THP-1 macrophages were transfected with the different concentrations of plasmid encoding E (I) or NS3 (J) for 24 h. Supernatants were analyzed (top) by gelatin zymography assays for MMP-9 proteinase activity. Cell lysates were analyzed (bottom) by immunoblotting. (**K**) PMA-differentiated THP-1 macrophages were transfected with the same concentrations of plasmid encoding NS1, E, or NS3 for 24 h. Supernatants were analyzed (top) by gelatin zymography assays for MMP-9 proteinase activity. Cell lysates were analyzed (bottom) by immunoblotting. Dates were representative of two to three independent experiments. Values are

mean ± SEM, P ≤0.05 (\*), P ≤0.01 (\*\*), P ≤0.001 (\*\*\*).
(TIF)

**S3 Fig. DENV2 induces MMP-9 expression and secretion in human PBMCs.** (**A**, **B**) Human
PBMCs were infected with DENV2 for different times at MOI = 5 (A) or at different concentrations for 24 h (B). Intracellular *MMP-9* RNA (top) and DENV2 *E* RNA (bottom) was determined by qRT-PCR analysis and MMP-9 proteinase activity in the supernatants was
determined by gelatin zymography assays (middle). Dates were representative of two to three
independent experiments. Values are mean ± SEM, P ≤0.05 (\*), P ≤0.01 (\*\*), P ≤0.001 (\*\*\*).
(TIF)

**S4 Fig. Detection of DENV infection in mice.** (**A**, **B**) *IFNAR*$^{-/-}$ C57BL/6 mice were intravenously injected with 300 μl DENV2 at a dose of $1\times10^6$ PFU/mouse (n = 6), pre-treated with
300 μl PBS containing MMP-9 specific inhibitor SB-3CT (5 mg/kg per mice) by intraperitoneal
injection for 90 min and then treated with DENV2 ($1\times10^6$ PFU/mouse), repeat treated with
SB-3CT (5 mg/kg per mice) on the fourth day after DENV2 (NGC) infection (n = 6), or 300 μl
PBS containing the same volume DMSO as a control group (n = 4). 7 days after infection,
mice were euthanasia, and the tissues were collected. Blood samples were collected at 2, 4, and
6 days post-infection. DENV2 *E* (A) and *NS5* (B) RNA was determined by qRT-PCR. Points
represent the value of each blood samples. Dates were representative of two independent
experiments. ns means not significant. P ≤0.05 (\*), P ≤0.01 (\*\*), P ≤0.001 (\*\*\*).
(TIF)

**S5 Fig. NS1 through recruiting MMP-9 to destroy the junctional molecules.** (**A**) Hela cells
were respectively transfected with plasmid encoding *MMP-9* (2 μg) or *NS1* (2 μg) or *NS1* (1
ug) plus *MMP-9* (1 μg) for 24 h or firstly co-transfected with plasmid encoding *NS1* (1ug) plus
*MMP-9* (1 μg) for 12 h, then treated with 600nM SB-3CT for 12 h. The indicated proteins in
cell extract were analyzed by WB. (**B–E**) Hela cells were treated with NS1 protein (5 μg/ml) or
MMP-9 protein (100 ng/ml) or NS1 (5 μg/ml) plus MMP-9 (100 ng/ml) or pre-incubated with
600nM SB-3CT for 1 h, then treated with NS1 (5 μg/ml) plus MMP-9 (100 ng/ml) for 6 h, The
distribution of endogenous β-catenin (B) or ZO-1 (D) protein were visualized under confocal
microscope. The quantification of relative β-catenin (C) or ZO-1 (E) intensity was used by
ImageJ software. Dates were representative of three independent experiments. The quantification of protein was used by ImageJ software (A).
(TIF)

**S6 Fig. NS1 interacted with junctional molecules, but MMP-9 have no interaction with the
junctional molecules.** (**A**, **C**) HEK293T cells were transfected with plasmid encoding *HA-NS1*
plus *Flag-β-catenin* (A) or *HA-NS1* plus *Flag-ZO-1* (C). Cell lysates were immunoprecipitated
using anti-Flag antibody, and analyzed using anti-Flag, anti-HA antibody. Cell lysates (40 μg)
was used as Input. (**B**, **D**) Hela cells were transfected with plasmid encoding *HA-NS1* (B) or
*HA-NS1* plus *Flag-ZO-1* (D). Cell lysates were immunoprecipitated using anti-β-catenin antibody (B) or anti-Flag (D), and analyzed using anti-Flag, anti-HA antibody and anti-β-catenin
antibody. Cell lysates (40 μg) was used as Input. (**E–H**) HEK293T cells (E, F) or Hela cells (G,
H) were transfected with plasmid encoding *Flag-MMP-9*. Cell lysates were immunoprecipitated using anti-Flag or anti-β-catenin antibody, and analyzed using anti-Flag or anti-β-catenin antibody. Cell lysates (40 μg) was used as Input. (**I**, **J**) HEK293T cells (I) or Hela cells (J)
were transfected with plasmid encoding *HA-MMP-9* plus *Flag-ZO-1*, Cell lysates were immunoprecipitated using anti-HA or anti-Flag antibody, and analyzed using anti-Flag or anti-
MMP-9 antibody. Cell lysates (40 μg) was used as Input. All dates were representative of three

independent experiments.
(TIF)

**S7 Fig. Detection of interaction domain of NS1 with MMP-9 or β-catenin / ZO-1.** (**A**) Schematic diagram of wild-type NS1 protein and truncated mutants NS1 protein (NS1-1 to NS1-8). (**B**) HEK293T cells were co-transfected with HA-MMP-9 and Flag-NS1 and its truncated mutants (NS1-1 to NS1-2). Cell lysates were immunoprecipitataed using anti-Flag antibody, and analyzed using anti-Flag and anti-HA antibody. Cell lysates (40 μg) was used as Input. (**C**) HEK293T cells were transfected Flag-NS1 and its truncated mutants (NS1-1 to NS1-2). Cell lysates were immunoprecipitataed using anti-Flag antibody, and analyzed using anti-Flag, anti-b-catenin, and anti-ZO-1 antibodies. Cell lysates (40 μg) was used as Input. (**D**) Schematic diagram of wild-type NS1 protein and truncated different domains of NS1 protein (NS1-D1 to NS1-D3). (**E**) HEK293T cells were co-transfected with HA-MMP-9 and Flag-NS1 and its truncated domains (NS1-D1 to NS1-D3). Cell lysates were immunoprecipitataed using anti-Flag antibody, and analyzed using anti-Flag and anti-MMP-9 antibody. Cell lysates (40 μg) was used as Input. (**F**) HEK293T cells were transfected Flag-NS1 and its truncated domains (NS1-D1 to NS1-D3). Cell lysates were immunoprecipitataed using anti-Flag antibody, and analyzed using anti-Flag, anti-β-catenin, and anti-ZO-1 antibodies. Cell lysates (40 μg) was used as Input. All dates were representative of three independent experiments.
(TIF)

**S8 Fig. Detection of MMP-9 knockout mice.** C57BL/6 mice and MMP-9$^{-/-}$ mice were injected intravenously DENV2 NS1 protein [10 mg/kg (n = 5)], the same volume of PBS was also tail vein injected to C57BL/6 mice and MMP-9$^{-/-}$ mice (n = 5) as control group. Another group of MMP-9$^{-/-}$ mice (n = 5) were injected intravenously DENV2 NS1 protein (10 mg/kg) plus recombinant mouse MMP-9 protein (70 μg/kg). After 24 h post-injection. The tails randomly selected from five groups (Four mice came from wild type C57BL/6 and twelve mice came from MMP-9$^{-/-}$ mice), the total genome was extracted from the tail of mice. The knock-out level of MMP-9 was detected by specific primers.
(TIF)

**S1 Table. Characteristics of sever dengue patients.**
(XLSX)

**S2 Table. The expression of NS1 and MMP-9 in individual dengue patient.**
(XLSX)

**S3 Table. qRT-PCR Primers used in this study.**
(XLSX)

**S4 Table. Primers used for plasmids construction in this study.**
(XLSX)

## Acknowledgments

We thank Dr. Jincun Zhao of the First Affiliated Hospital of Guangzhou Medical University, Guangzhou, China, for the gift of IFNAR$^{-/-}$C57BL/6 mice deficient in IFN-α/β receptors, Dr. Xulin Chen of Wuhan Institute of Virology, Chinese Academy of Sciences, for the gift of DENV-2 strain NGC (GenBank accession number KM204118.1).

## Author Contributions

**Conceptualization:** Pan Pan, Geng Li, Luping Lin, Jianguo Wu.

**Data curation:** Pan Pan, Weiwei Ge, Keli Chen, Muhammad Adnan Shereen, Zhen Luo, Xulin Chen, Qiwei Zhang.

**Formal analysis:** Pan Pan, Miaomiao Shen, Zhenyang Yu, Zizhao Lao, Yaohua Fan, Keli Chen, Zhihao Ding, Wenbiao Wang, Pin Wan, Luping Lin.

**Funding acquisition:** Jianguo Wu.

**Investigation:** Pan Pan, Geng Li, Zhenyang Yu, Weiwei Ge, Keli Chen, Wenbiao Wang, Pin Wan, Muhammad Adnan Shereen, Zhen Luo, Luping Lin.

**Methodology:** Pan Pan, Geng Li, Miaomiao Shen, Zhenyang Yu, Weiwei Ge, Zizhao Lao, Yaohua Fan, Zhihao Ding, Wenbiao Wang, Muhammad Adnan Shereen, Xulin Chen, Qiwei Zhang.

**Project administration:** Jianguo Wu.

**Resources:** Pan Pan, Geng Li, Miaomiao Shen, Zhenyang Yu, Weiwei Ge, Zizhao Lao, Yaohua Fan, Zhihao Ding, Pin Wan, Zhen Luo, Xulin Chen, Qiwei Zhang, Luping Lin, Jianguo Wu.

**Supervision:** Jianguo Wu.

**Validation:** Pan Pan, Geng Li, Miaomiao Shen, Zhenyang Yu, Weiwei Ge, Zizhao Lao, Zhihao Ding, Wenbiao Wang, Pin Wan, Muhammad Adnan Shereen, Zhen Luo, Luping Lin.

**Visualization:** Yaohua Fan, Keli Chen, Wenbiao Wang, Xulin Chen, Qiwei Zhang, Jianguo Wu.

**Writing – original draft:** Pan Pan, Geng Li, Luping Lin, Jianguo Wu.

**Writing – review & editing:** Geng Li, Jianguo Wu.

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
