## [Decision Letter · Decision Letter 0]

4 Jul 2020

Dear Dr. Wu,

Thank you very much for submitting your manuscript "DENV NS1 and MMP-9 cooperate to induce vascular leakage by altering endothelial cell adhesion and tight junction" for consideration at PLOS Pathogens. As with all papers reviewed by the journal, your manuscript was reviewed by members of the editorial board and by several independent reviewers. In light of the reviews (below this email), we would like to invite the resubmission of a significantly-revised version that takes into account the reviewers' comments with new experiments. Some major concerns were raised about the cellular localization of transfected NS1 and its relationship to virus-produced NS1. These would need to be addressed adequately for us to proceed.

We cannot make any decision about publication until we have seen the revised manuscript and your response to the reviewers' comments. Your revised manuscript is also likely to be sent to reviewers for further evaluation.

Sincerely,

Mehul Suthar

Associate Editor

PLOS Pathogens

Michael Diamond

Section Editor

PLOS Pathogens

Kasturi Haldar

Editor-in-Chief

PLOS Pathogens

orcid.org/0000-0001-5065-158X

Michael Malim

Editor-in-Chief

PLOS Pathogens

orcid.org/0000-0002-7699-2064

Reviewer's Responses to Questions

**Part I - Summary**

Reviewer #1: In this paper, Wu and colleagues report in vitro and in vivo studies on the effect of DENV NS1 protein and host protein matrix metalloproteinase-9 (MMP-9) on permeability of endothelial cells. An important function of DENV NS1 in decreasing endothelium permeability and severe dengue pathogenesis has been established. Previous reports indicate that DENV NS1 disrupts the endothelial glycocalyx layer to compromise the integrity of the vascular permeability barrier. It was also shown previously that DENV-infected dendritic cells trigger vascular leakage through MMP-9 overproduction. Here the authors link NS1 to MMP-9 overproduction and present the following arguments in this paper: 1) DENV NS1 transcriptionally increase MMP-9 expression via NFkB activation; 2) DENV NS1 inhibits TIMP-1 production; 3) DENV NS1 interacts with both MMP-9, ZO-1, and beta-catenin, recruiting MMP-9 to these junction proteins to facilitate the deterioration of both adhesion and tight junctions. Although the study provides strong data confirming the role of both NS1 and MMP-9 in DENV-induced vascular leakage, the mechanistic parts, which are the novel parts of the study, has drawbacks and appear preliminary at this stage.

Reviewer #2: Vascular leakage and hemorrhage are key features of severe dengue disease, so understanding the viral mechanisms that trigger endothelial permeability is important. Prior work has shown that the viral NS1 protein can promote endothelial permeability, but the mechanisms by which this occurs are not entirely clear. Here, Li and colleagues characterize a coordinated effect of NS1 and MMP9 on cell junction molecules and ultimately endothelial barrier permeability. The study is comprehensive, including patient samples, mouse models, and molecular studies. In general the experiments are thorough but additional clarifications would allow the authors to draw more robust conclusions.

A weakness of the proposed mechanism is that the interaction experiments are all done in cells transfected with tagged NS1 and MMP9, rather than during infection; this may or may not represent the authentic localization of NS1 during infection, especially since NS1 is a secreted protein. The authors note cytoplasmic localization of NS1 in their transfection system (Fig 2, Line 171-172, Fig 9), but during infection NS1 is present in the ER lumen and secreted. Is NS1 secreted in this system? Do the authors propose that NS1 and MMP9 interact in the cytoplasm?

Fig 1: If patient sera are still available, it would be helpful to show viremia by qRT-PCR. Is high NS1 per se important, or is high NS1 a marker for high viral load, which could impact vascular permeability by other mechanisms?

Line 157: The correlation between TIMP1 and NS1 is not compelling.

For qRT-PCR, it is not clear what expression levels are relative to (if they are already normalized to GAPDH). (Fig 3, Fig 4)

Fig 4E: Why is there a single mock for both cell types? If relative to GAPDH, these shouldn't be averaged together. It would be better to show viral loads based on a standard curve, on a Log10 scale, rather than relative to GAPDH.

Fig 5B: Is it surprising that the MMP9 inhibitor reduces MMP9 levels? Does that suggest that this drug acts by blocking MMP9 production, rather than inhibiting its activity?

Fig 5I: Hyperpermeability induced by NS1 alone is observed at only a single timepoint (7hrs), whereas the MMP9 treatment induces continual decrease in TEER starting at 3hrs. Adding MMP9 + NS1 had no effect greater than MMP9 alone. Altogether these data suggest that the hyperpermeability effect is entirely due to MMP9, without a contribution from NS1. (Line 272-273)

Fig 6A-C, Fig 8D-I: The effect DENV infection or NS1 on expression of cell junction proteins in mice does not obviously match the authors’ description (line 282-285, 334-341), e.g. ZO-2 in liver, b-catenin in spleen. This is not necessarily surprising, it would require a very strong effect to downregulate the total expression of these proteins in a full tissue.

Reviewer #3: In the manuscript by Li and colleagues entitled, “DENV NS1 and MMP-9 cooperate to induce vascular leakage by altering endothelial cell adhesion and tight junction”, the authors describe potential mechanisms by which NS1 induces vascular leak that involve increased expression and activation of the matrix metalloprotease 9 (MMP-9) via NF-κB signaling pathways after DENV infection of human primary PBMCs or human monocytic cell lines (e.g. THP-1) in vitro. The authors also performed a series of experimental assays in vitro and in vivo to suggest that NS1 may interact with MMP-9 to facilitate disruption in the integrity of tight junctions, leading to endothelial hyperpermeability and vascular leakage in mouse tissues. This effect was inhibited in vitro by the specific inhibitor of MMP-9 SB-3CT and was relatively unaffected by NS1 in MMP9-/- mice, which suggests that MMP-9 is required for NS1-induced hyperpermeability.

The fact that the NS1 protein of flaviviruses can modulate the expression and activity of MMP-9 to perturb the integrity of biological barriers has been the focus of different groups, who have shown the effect of NS1 alone or NS1 in the context of viral infection (e.g., ZIKV) on the permeability of distinct biological systems such as the blood-testis barrier or the endothelial microvasculature (Chen et al., 2018; Hui et al., 2020).

In the present study, authors present a significant set of biological and biochemical data explaining the potential NS1-induced mechanism leading to this MMP-9 phenomenon and its consequences on endothelial barrier function. Although the authors clearly demonstrate the requirement for MMP-9 in NS1-mediated endothelial dysfunction in vivo and in vitro (also shown by others), many of the mechanistic assertions from this study are not justified by the data. Therefore, certain additional experiments are required and questions and technical issues must be addressed, as outlined below in the sections that follow.

**Part II – Major Issues: Key Experiments Required for Acceptance**

Reviewer #1: 1) For argument one, it’s not clear how NS1 triggers NFkB activation. The experimental data was obtained from transfection experiments, it’s not clear whether NS1 is secreted in this system, and if not the secreted form (as suggested by the model), how would an ER-resident form of NS1 activate the NFkB pathway? NS1 has previously shown to activate other cytokines, is the mechanism here expected to be different?

2) For argument two, how does NS1 decrease TIMP-1 production while simultaneously increase MMP-9 transcription? Isn’t TIMP-1 also a target gene for NFkB activation? Is the TIMP-1 reduction at the transcriptional, translation, or secretion level?

3) For argument three, where do all these interactions supposed to occur to be relevant for the observed biological effect? The co-IP and co-staining are all intracellular, but the argument appears to be the secreted NS1 would be mediating effect on ECs in vivo as ECs are not readily infected. So the relevant interaction would be on the extracellular side. There is no data showing that extracellular NS1 having these interactions. Given that the form of the intracellular NS1 monomer/dimer is different from the extracellular hexamer, the binding data from the transfection experiments may not be relevant for pathogenesis in vivo. In Fig. 9, it’s not clear how the extracellular NS1-MMP-9 complex would have access to ZO-1.

4) For argument three, if NS1 were to bridge an interaction between MMP-9 and the junction proteins, one would expect them to bind to different parts of NS1. This was not investigated. What about any mutant of NS1 that have been reported to lose the ability to cause EC hyperpermeability? Does the mutation affect any of the phenotypes reported here?

5) In Fig. 6, the authors show that beta-catenin and the TJ proteins can be downregulated by MMP-9 alone, and to a lesser extent by NS1 alone. In fact, adding NS1 in this assay did not show more downregulation than MMP-9 alone. Then in Figure 7, the authors show that MMP-9 in unable to bind these junction proteins unless NS1 is present.

Reviewer #3: 1. In Figure 1, the authors examined the levels of NS1, MMP-9, and TIMP-1 circulating in the sera of DENV-infected patients and healthy donors, finding that the levels of soluble NS1 were increased from day 2 until day 11, the last day of sample collection; however, the levels of MMP-9 seem to be significantly increased after 11 days. In patients experiencing severe dengue, increased vascular leak typically becomes clinically evident after 4-6 days of onset of symptoms, known as the critical phase. Can authors explain this apparent discordance in the timeline with respect to the appearance of these viral and host markers and the biological effect proposed by the study?

2. In Figure 2, authors assessed the potential interaction between DENV protein(s), including NS1, and MMP-9 in co-transfected 293T cells by performing a co-immunoprecipitation (Co-IP) assay using anti-Flag antibody. Authors should perform Co-IP experiments using either an anti-NS1 or anti-MMP9 mAb. Is this pulled-down MMP-9 more active after interaction of NS1?

3. The authors claim that an interaction between NS1 and MMP9 results in NS1-mediated TJ/AJ disruption, and while the data are certainly suggestive, they are certainly not conclusive. To make this assertion, the authors must demonstrate that disrupting this interaction ablates NS1-mediated pathology; this can be done using mutants of either NS1 or MMP9.

4. Further, the authors claim that TIMP-1 may be playing a role in regulating MMP-9 activity. How do the authors propose this to be the case? Either way, the authors do not functionally demonstrate a relationship between TIMP1/NS1 and MMP9. If the authors want to assert this point, they must provide further experiments.

5. The IFA in Figure 2C displays blown-out cytoplasmic signal and thus no conclusion of colocalization can be made. Please present an image with less blown-out signal.

6. It is well appreciated that Co-IP assays have a propensity to yield false positives, especially when overexpressing proteins in 293T cells. The authors should demonstrate interaction of endogenous proteins in infected THP1 or other cells (ideally, the authors can demonstrate an interaction by pulling down either protein). Further, it would be best to demonstrate the interaction in an orthogonal assay. On the same point, IFA colocalization studies between all proteins reported to interact would be helpful in interpreting the data.

7. All Western blot data showed in Figure 2 correspond to anti-HA and anti-Flag antibodies. As Pro- vs Active MMP-9 have different molecular weights, what is the molecular weight of the MMP-9 shown in Figure 2? And in the other Figures throughout the manuscript?

8. In Figure 4, the authors describe a series of data regarding how DENV2 induces MMP-9 expression and secretion in human PBMCs and macrophages but not in HUVECs. DENV infection of human endothelial cells has been always a controversial topic, as these cells are not considered a target for DENV replication in vivo as monocytes and other immune cells are. Interestingly, here the data showed that indeed HUVECs are not significantly infected with DENV during the timeline of the experiment (24 hours) and thus, changes in the expression of MMP-9 were not detected. Along these lines, can the authors overexpress NS1 in HUVECs (or treat with secreted NS1) to assess whether this alters MMP-9 expression?

9. What were the levels of NS1 expression on THP-1 cells infected with DENV? This piece of data is missing, and it is important to show that levels of NS1 may correlate with the levels of MMP-9 expression in infected cells as shown in Figure 3A, where THP-1 cells were transfected with NS1.

10. In Figure 5, the authors showed that supernatants collected from DENV-infected THP-1 cells increased the permeability of naïve HUVEC monolayers, which was prevented by an MMP-9-specific inhibitor (SB-3CT). There are several points that need to be clarified here: 1) Did authors measure the levels of NS1 present in the supernatant collected from DENV-infected THP-1 cells that were added on top of HUVEC monolayers? This is important to specify, as soluble NS1 can also modulate the permeability of HUVEC monolayers. 2) The authors must use an irrelevant chemical inhibitor to demonstrate the specificity of the TEER inhibition showed in Figure 5A once SB-3CT was mixed with the infected supernatant.

11. In Figure 5B, the authors examined the effect of the MMP-9-specific inhibitor SB-3CT in IFNAR-/- mice infected with DENV2 and then measured, among other things, the amount of MMP-9 circulating in the sera of the experimental mice. Do authors know if these types of inhibitors may have pleiotropic effects in vivo that may have affected virus replication? Also, what were the viremia levels in DENV-infected mice treated and not treated with this inhibitor? This is not shown in Figure 5.

12. As suggested above for Figure 5A, authors must include irrelevant (unrelated) chemical inhibitor controls in these TEER experiments (Figure 5I).

13. In Figure 6A-C and G, the Western blot data is exceedingly difficult to interpret as bands corresponding to tight or adherens junction proteins look shortened and sometimes fussy. Authors should repeat these Western blots to show better images and include a panel with quantitative analysis of the protein expression.

14. Figure 6D-F, in which authors aim to show the expression/staining of beta-catenin, are not interpretable. Authors should perform some type of protein quantification.

15. In Figure 8A-C, authors show very nicely the effect of NS1, MMP-9 and the knockout MMP-9-/- on vascular leak detected in mice. However, again the Western blot data in panels E and F are not easy to interpret, particularly for ZO-2 and beta-catenin proteins. The authors should repeat this and show quantitative analysis of the protein expression.

**Part III – Minor Issues: Editorial and Data Presentation Modifications**

Reviewer #1: 1) This group previously published a similar study on DENV M Protein, which they show that it can induce vascular leakage in mice by interacting with the NLRP3 Inflammasome. Animal model data was also presented in that paper. That study doesn’t appear to be cited or discussed here. Which mechanism do the authors believe is more important?

2) There is an issue of missing controls and data quantification. Are the NFkB activation or MMP-9 bindings specific to DENV NS1 or other flavivirus NS1 can also do this? I think it would be important to investigate this if conclusions were to be drawn for dengue pathogenesis. For NKkB activation, what about other targets or non-target genes?

3) Fig. 5I. TEER values should be shown. These values for HUVEC are typically very low to begin with, a 10-20% reduction may not be that relevant for barrier function.

4) Fig. 8D. ZO-2 is similarly downregulated by NS1 regardless of MMP-9 was knocked out or not.

5) The paragraph title: “DENV enhances the production of MMP9 and TIMP-1 in severe dengue patients” is incorrect. TIMP-1 production decreases in patients as well as in vitro with NS1 expression.

6) Line 157, “converse correlation” is incorrect, should be “inverse correlation”

7) Line 308, “failed to interaction with” should be “failed to interact with”.

Reviewer #2: Line 51-52: the claim that the proposed mechanism is “the main reason leading to death in severe dengue patients” should be toned down.

Line 105-106: As written this implies that facilitating tumor migration is the normal physiologic function of MMP9

Fig 1: For D,E,F, re-orient graphs so independent variable (D NS1, E TIMP1, F NS1) is on the X axis and dependent variable on the y axis.

The manuscript would benefit from another round of editing to polish the writing.

Reviewer #3: 1. There are many grammatical errors and typos throughout the manuscript. Please revise carefully.

2. Could the authors specify which serotype of DENV NS1 is purchased from the Native Antigen Company?

3. The MMP9/NS1 connectionis not a new finding, please cite relevant literature to describe what is reported previously (2-3 papers at least).

4. Author summary line 51-52: The authors write, “We found that a distinct mechanism of DENV NS1 and MMP-9 cooperatively induce vascular leakage is the main reason leading to death in severe dengue patients.” This statement is a wild overstatement that is not supported by the data. It is also likely not accurate, as vascular leakage in severe dengue disease is undoubtedly multifactorial and complex.

5. Figure 4E, label on y-axis is missing.

6. Figure 5B, label on y-axis is missing.

7. Figure 5F, “live” instead of “liver”.

PLOS authors have the option to publish the peer review history of their article (what does this mean?). If published, this will include your full peer review and any attached files.

Reviewer #1: No

Reviewer #2: No

Reviewer #3: No
---

## [Decision Letter · Decision Letter 1]

9 Dec 2020

Dear Dr. Wu,

Thank you very much for submitting your manuscript "DENV NS1 and MMP-9 cooperate to induce vascular leakage by altering endothelial cell adhesion and tight junction" for consideration at PLOS Pathogens. As with all papers reviewed by the journal, your manuscript was reviewed by members of the editorial board and by several independent reviewers. In light of the reviews (below this email), we would like to invite the resubmission of a significantly-revised version that takes into account the reviewers' comments. However, this next version will be the final one, as we do not want to continue an iterative process of review, so please address the reviewer concerns (especially #3 and #4) carefully.

We cannot make any decision about publication until we have seen the revised manuscript and your response to the reviewers' comments. Your revised manuscript is also likely to be sent to reviewers for further evaluation.

Sincerely,

Mehul Suthar

Associate Editor

PLOS Pathogens

Michael Diamond

Section Editor

PLOS Pathogens

Kasturi Haldar

Editor-in-Chief

PLOS Pathogens

orcid.org/0000-0001-5065-158X

Michael Malim

Editor-in-Chief

PLOS Pathogens

orcid.org/0000-0002-7699-2064

Reviewer's Responses to Questions

**Part I - Summary**

Reviewer #2: In this revised manuscript, Pan and colleagues have made revisions and added new data to respond to reviewer comments. Many reviewer concerns have been addressed.

Reviewer #3: This revised manuscript by Pan et al., addresses some of my original comments, but a number of critical questions remain. As the functional relevance of MMP9 in NS1-mediated vascular leak has been previously reported, the novel elements of this manuscript are the mechanistic interaction data. As such, the most critical points the authors still need to address include demonstrating the functional relevance of the interactions between NS1 and MMP9 and demonstrating the protein-protein interactions (between NS1 and MMP9) using an orthogonal approach to the current co-immunoprecipitation assays, which are not fully convincing on their own. Further, although the correlations are interesting, the authors’ assertion that TIMP1 plays a role in their model must be functionally validated or else should be de-emphasized. The authors should address the points below.

Reviewer #4: The manuscript attempts to define how NS1 induces EC permeability in dengue infections. The authors have identified MMP-9 as an important modulatory of permeability by NS1. This is important. Information but has been shown previously. The mechanistic data as to how NS1 does this uses several different cells lines which makes the manuscript confusing and fails to account for changes in structure of the form of NS1 both inside and outside the cell. There is some concern that the HUVEC data are not robust as the authors appear to have not created tight barriers to address the questions.

**Part II – Major Issues: Key Experiments Required for Acceptance**

Reviewer #3: 1. MMPs are secreted as inactive proenzymes and are activated by the cleavage of the propeptide domain by proteolytic enzymes. In this study, one of the main conclusions suggested by the authors is that NS1 interacts with the MMP9 expressed in immune cells, leading to degradation of tight and adherens junction proteins on endothelial cells. However, what is the result of this interaction? Does NS1 modulate the proteolytic activity of MMP9, leading to its activation? Authors must experimentally demonstrate this phenomenon.

2. Most importantly, regarding the new data in fig. S4B, MMP9 truncation mutants cannot be used to make a functional conclusion regarding whether the interaction between MMP9 and NS1 is critical for NS1-mediated cell-cell junction disruption, as the loss of this phenotype can be explained either by the loss of interaction between NS1 and MMP9 or simply by the loss of protein function due to MMP9 truncation. The authors would need to create point mutants of MMP9 and/or NS1 that no longer interact and demonstrate that these mutants do not mediate intercellular junction disruption.

3. The data in Figure 7J are confusing, as it appears that all truncation mutants of NS1 interact with MMP9, suggesting that this interaction may not be specific, which is not uncommon in CO-IP assays. Can the authors explain these data?

4. To confirm the specificity of the interaction of NS1 with MMP9, the authors should utilize an orthogonal method (e.g., Biomolecular fluorescence complementation (BIFC), yeast 2-hybrid, or through convincing subcellular colocalization).

5. If the proposed model is correct, the authors should be able to visualize NS1 and MMP9 colocalizing at cell-cell junctions when added to cells exogenously. Such an observation is critical to support the model proposed by the authors. To clarify, the overexpression data in Fig. 2D demonstrate broad cytoplasmic localization of overexpressed proteins and not colocalization (as the signal of both proteins is everywhere in the cytoplasm).

6. As no function is demonstrated for TIMP1, the authors need to experimentally demonstrate why this is important or remove from their central mechanistic message (refer to the association result).

Reviewer #4: In figure 1. The authors state they are severe dengue patents. How is severity classified? Were they all DHF or DSS patients? The authors should include both primary and secondary dengue patients and both severe and non-severe to truly understand how NS1 is modulating the level of TIMP-1 and MMP-9. If MMP-9 and TIMP-1 are important for severity they should not be changed in non-severe dengue.

Furthermore, given that the clinical samples had both IgG and IgM levels, classification of primary or secondary is possible based on this data. As is the determination if NS1 increases over time due to an increase in virus levels (as we know it does), as the methods state that qPCR for DENV viral load was performed and the data are then available. Contrary to the response to Reviewer 2 when asked for this data. These data need to be included.

It would also be useful to have each individual patient marked with a different symbol so that they can be followed specifically in A B and C. Do all patients show this increase? Or only some? Is there a correlation to viral load?

In figure 3. Given that Dengue does not infect endothelial cells in vivo, expressing NS1 in HUVECs to determine if it is secreted is not relevant.

There is no information in the methods detailing how HUVECs were transfected, or indeed any of the cells used in the study. HUVECs are incredibly difficult to transfect and keep alive and this information needs to be included.

There is no discussion on why HUVECs expressing NS1 did not affect MMP-9 expression.

In figure 4 the authors evaluate the infection level in THP-1 and HUVECs. They demonstrate increased viral copies in THP-1 and that infection increases with dose of dengue. This experiment needed a UV-inactivated and No RT control to ensure the data is detecting de novo E and not input virus genome.

In Fig 5. The authors evaluate EC permeability in response to supernatants derived from dengue infected HUVECs or THP-1 cells. There is no indication in the TEER assay that the HUVECs were allowed to establish a tight barrier. The methods do not indicate how many cells were seeded and how long they were left to form barriers. 5J also demonstrates that NS1 alone induced permeability at 7 h post-treatment. This is considerably different than what has been published by both the Harris and King groups. Please explain. And how does the data fit with the lack of MMP-9 generated in HUVECs in response to NS1? What is the factor here that is inducing the permeability to the same levels as NS1 alone? These data do not seem to support the role of MMP-9 as a central mediator but rather simply an additional mechanism that could induce permeability.

In the immunofluorescent experiments (Fig 6)the methods state that HUVECs were seeded onto coverslips and grown to 80% confluency and then treated with NS1, MMP-9 or both for 6 h. Cells were then subsequently stained for tight junction proteins. This is not the correct way to do this experiment. Endothelial cells, including HUVECs, require approximately 4-5 days to establish 100% confluency and develop true tight junctions. This experiment should be repeated with HUVECs that have been allowed to form tight junctions creating a tight barrier.

In figure 7. The authors use truncated NS1 to determine the interaction domains with tight junction proteins. Given soluble NS1 is a hexamer, and in the cell NS1 exists as a dimer, the authors need to determine how the truncations affect the structure, as this may count for the inability to interact.

The data are slightly confusing as to which form of NS1 is mediating what interaction. For the transfection experiments and MMP-9 interaction, this would be dimeric NS1 interacting in the cell. For the modification of tight junction proteins with the addition of NS1 externally, this would suggest the hexamer is mediating the effect. How are the domains that are indicated modeled on the hexamer? Are the on the outside available for interaction or is the interaction abrogated simply because the structure is destroyed?

**Part III – Minor Issues: Editorial and Data Presentation Modifications**

Reviewer #2: Fig 1E,F: The TIMP-1 correlations are not very compelling

Fig 2: It is unclear how the experiment in Fig 2B differs from the one in Fig 2A.

Fig 2D: This co-localization does not seem very robust.

Are the NS1 blots in Fig3F and 3G correct?

Fig 4E: The revisions do not address the original question: how is there a single mock for 2 different cell types, with the same number of replicates as the infected samples? Either this is from THP-1 or from HUVEC, but it is not indicated. It doesn’t really matter (this mock doesn’t even need to be included in the figure), but the current presentation doesn’t make sense.

Fig 5J: The data do not support the conclusion that adding NS1 increased endothelial permeability compared to adding MMP9 alone.

Fig 7: Check sample designations in 7E and F.

The interaction phenotypes of the truncation mutants don’t make a lot of sense. It also seems that these truncations were not designed with particular consideration of known NS1 domains.

Reviewer #3: 1. Overall, it seems as if MMP9-mediated intercellular junction disruption can occur independently from NS1, which is inconsistent with the model proposed by the authors (that NS1 links the interaction between TJ/AJ proteins and MMP9). How can the authors explain this?

2. The authors should describe in further detail the proposed mechanisms previously reported for MMP9 in NS1-mediated pathogenesis (e.g., Chen et al., 2019), as this pathway has been studied previously. It would be helpful to contrast this with the proposed mechanism from this study.

3. The authors need to add scale bars to all IFA images.

4. In Fig. S4C, the panels for NS1 and MMP9 appear to be the same panels with different signal intensity (these are duplicated panels), indicating a problem with figure design.

5. Lines 302 and 303: this is not the first study to reveal TJ/AJ disruption, please include (https://doi.org/10.1099/jgv.0.001163) and (https://doi.org/10.1099/jgv.0.001401)

6. Line 142: “opt” should be “cooperate”.

7. Line 203, do authors mean NS1-induced MMP9 expression?

8. Lines 310-311, this statement is not justified by the data.

PLOS authors have the option to publish the peer review history of their article (what does this mean?). If published, this will include your full peer review and any attached files.

Reviewer #2: No

Reviewer #3: No

Reviewer #4: No
---

## [Decision Letter · Decision Letter 2]

16 Mar 2021

Dear Dr. Wu,

Thank you very much for submitting your manuscript "DENV NS1 and MMP-9 cooperate to induce vascular leakage by altering endothelial cell adhesion and tight junction" for consideration at PLOS Pathogens. As with all papers reviewed by the journal, your manuscript was reviewed by members of the editorial board and by several independent reviewers. In light of the reviews (below this email), we would like to invite the resubmission of a significantly-revised version that takes into account the reviewers' comments.

One of the reviewers feels that several key experiments are lacking to support the central hypothesis. The authors should pay particular attention to the inclusion of appropriate controls (as outlined by the reviewer), performing mutational analysis to further demonstrate the interactions between MMP9 and NS1, clarification on Fig 3H, and demonstrating co-localization of NS1 and MMP9 with TJ/AJ of cells.

We cannot make any decision about publication until we have seen the revised manuscript and your response to the reviewers' comments. Your revised manuscript is also likely to be sent to reviewers for further evaluation.

Sincerely,

Mehul Suthar

Associate Editor

PLOS Pathogens

Michael Diamond

Section Editor

PLOS Pathogens

Kasturi Haldar

Editor-in-Chief

PLOS Pathogens

orcid.org/0000-0001-5065-158X

Michael Malim

Editor-in-Chief

PLOS Pathogens

orcid.org/0000-0002-7699-2064

Reviewer's Responses to Questions

**Part I - Summary**

Reviewer #2: The authors have added additional data and responded to reviewer comments.

Reviewer #3: This revised manuscript by Pan et al., is improved but still lacks key experiments to support the authors central hypotheses of how an interaction between NS1 and MMP9 alters endothelial dysfunction. As stated in the previous review, a role for MMP9 in NS1-mediated endothelial dysfunction has been previously reported, and the advancement of this paper is that a direct protein-protein interaction between NS1 and MMP9 (1) modulates enzymatic activity of MMP9 and (2) localizes secreted MMP9 to the TJ/AJ of endothelial cells. While these conclusions of the authors are certainly reasonable, they are not yet fully supported by the data. The authors must address the following key points detailed below.

Reviewer #4: The authors have addressed many of the concerns I had.

**Part II – Major Issues: Key Experiments Required for Acceptance**

Reviewer #2: (No Response)

Reviewer #3: 1. While it is clear that a direct interaction between NS1 and MMP9 is occurring, the authors have not yet established that this is not simply a non-specific interaction owing to stickiness of MMP9 (specifically AA 440-707 according to their MMP9 truncation mutants). To test this, the authors require a proper negative control in their CO-IP, BiFC, and Y2H assays where a protein not interacting with MMP9 is successfully expressed/pulled down, and shown NOT to interact with MMP9; simply an empty vector where no protein is expressed is not an adequate control. The authors could take advantage of one of their DENV expression constructs, such as E, or any other nonstructural proteins, as tested in Fig. 2A, which appears to be in hand and would serve the purpose.

2. The authors need to address the function of this interaction, for example by: (1) repeating the gelatinase assay with 3 separate conditions using NS1, NS3, and E (these conditions would address whether gelatinase enhancement of MMP9 is specific to NS1, protein-protein interactions, or if it is non-specific); (2) making mutants of NS1 or MMP9 that no longer interact and demonstrate the lack of function with these proteins.

3. Regarding the gelatinase assay in Figure 3H, could the authors explain why the gelatinase activity of MMP9 increases in the presence of NS1 even though pro-MMP9 does not seem to mature to a lower band on the Western blot? How do the authors propose that NS1 increases MMP9 enzymatic activity? It doesn’t seem to be promoting maturation of pro-MMP9. Could the authors also display more of this Western blot, it appears that a lower band may be cut off?

4. Finally, since the authors claim that NS1 and MMP9 localize to the TJ/AJ of cells, this needs to be shown. Overexpressing TJ/AJ proteins in HeLa and 293T cells is not sufficient to demonstrate this point, as these proteins are clearly localized everywhere in the cytoplasm and are not forming actual junctional complexes. The best way to do this colocalization would be using a relevant model of polarized human endothelial cells such as HUVECs and detect endogenous TJ/AJ proteins (many good antibodies exist) and determine whether extracellular NS1 and MMP9 colocalize. If this could not be done, the authors would need to revise their overall model to exclude the steps not supported by data.

5. Finally, all IFA should be quantified in some way, especially the colocalization data (BiFC and colocalization figures), as it is important to demonstrate that the images are truly representative of the overall trend.

Reviewer #4: (No Response)

**Part III – Minor Issues: Editorial and Data Presentation Modifications**

Reviewer #2: Line 155-156: A correlation between NS1 and MMP9 levels in serum does not suggest anything about a mechanistic role for an interaction between the two, please rephrase or delete.

Line 197-199: It is unclear how low MMP9 expression in HUVEC is an explanation for the inability of NS1 to induce its expression. Fig 4G is called out of order in the text

Fig 9: change “adherent junction” to “adherens junction”

Reviewer #3: Could the authors add more detail to their Y2H protocol? Is the Figure correct with the dropout media combination (-TRP,-LEU,-HIS,-ADE), or is this the negative control condition? What about the -TRP,-LEU,-HIS condition? Clarifying the complementation combination and yeast strain would clarify this point.

Reviewer #4: Minor points

The authors stated that the new fig 1 had each individual patient as a different symbol as requested. This is not apparent on the figure.

Lines 186-190 do not make sense

In lines 251-253 the authors indicate that THP-1 cells induce high levels of MMP9 in response to dengue infection while HUVECs exhibited ‘modest’ elevation in MMP9. Looking at the presented data in Fig 4G there is no production of MMP9. The statement is not correct.

Line 214. Capitalize Results

PLOS authors have the option to publish the peer review history of their article (what does this mean?). If published, this will include your full peer review and any attached files.

Reviewer #2: No

Reviewer #3: No

Reviewer #4: No
---

## [Decision Letter · Decision Letter 3]

17 Jun 2021

Dear Dr. Wu,

Thank you very much for submitting your manuscript "DENV NS1 and MMP-9 cooperate to induce vascular leakage by altering endothelial cell adhesion and tight junction" for consideration at PLOS Pathogens. As with all papers reviewed by the journal, your manuscript was reviewed by members of the editorial board and by several independent reviewers. The reviewers appreciated the attention to an important topic. Based on the reviews, we are likely to accept this manuscript for publication, providing that you modify the manuscript according to the review recommendations. The authors should focus on addressing the comments by Reviewer #3.  In particular, this reviewer continues to have concerns about the lack of appropriate controls, subcellular localization of NS1/MMP9, and the functional relevance of this interaction. 

Sincerely,

Mehul Suthar

Associate Editor

PLOS Pathogens

Michael Diamond

Section Editor

PLOS Pathogens

Kasturi Haldar

Editor-in-Chief

PLOS Pathogens

orcid.org/0000-0001-5065-158X

Michael Malim

Editor-in-Chief

PLOS Pathogens

orcid.org/0000-0002-7699-2064

Reviewer Comments (if any, and for reference):

Reviewer's Responses to Questions

**Part I - Summary**

Reviewer #3: The authors have addressed some of my concerns, but issues remain. The authors have now sufficiently demonstrated that NS1 interacts specifically with MMP9 (although it is strange that all domains of NS1 interact with MMP9, suggesting that NS1 is a sticky protein). However, while the authors provide some evidence that NS1 promotes maturation of MMP9 and increased gelatinase activity, the functional relevance of this interaction remains unclear given the current data. Further, the subcellular localization and colocalization of NS1/MMP9 remain unclear, which is critical given the model drawn by the authors.

Reviewer #4: The authors have addressed my concerns.

**Part II – Major Issues: Key Experiments Required for Acceptance**

Reviewer #3: 1. The authors should discuss the caveats of their data, stating clearly that the functional relevance of the interaction between NS1 and MMP9 is not demonstrated in this paper.

2. The IFA in Figure 7K and L still lack controls: they require treatment of MMP9 and NS1 alone as well as in combination. Further, Fig. 7K does not show colocalization of MMP9 and NS1; the green and red stainings in the perinuclear region are clearly separate and not overlapping. Representative images need to show multiple, confluent cells (1) to demonstrate breakdown of intercellular junctions and (2) to demonstrate that co-localization occurs and is representative of the entire sample. Lastly, the localization appears to be intracellular (perinuclear), which is inconsistent with the author’s overall model. The NS1 staining pattern seems to be atypical of what is observed by others as well.

Reviewer #4: (No Response)

**Part III – Minor Issues: Editorial and Data Presentation Modifications**

Reviewer #3: The authors ideally should provide controls demonstrating expression of E and NS3 for their new figures (Figure 2C-D, Figure S1A-C and F).

Reviewer #4: (No Response)

PLOS authors have the option to publish the peer review history of their article (what does this mean?). If published, this will include your full peer review and any attached files.

Reviewer #3: No

Reviewer #4: No

Figure Files:

Data Requirements:

Reproducibility:

References:

---

## [Editor Report · Decision Letter 4]

6 Jul 2021

Dear Dr. Wu,

We are pleased to inform you that your manuscript 'DENV NS1 and MMP-9 cooperate to induce vascular leakage by altering endothelial cell adhesion and tight junction' has been provisionally accepted for publication in PLOS Pathogens.

Best regards,

Mehul Suthar

Associate Editor

PLOS Pathogens

Michael Diamond

Section Editor

PLOS Pathogens

Kasturi Haldar

Editor-in-Chief

PLOS Pathogens

orcid.org/0000-0001-5065-158X

Michael Malim

Editor-in-Chief

PLOS Pathogens

orcid.org/0000-0002-7699-2064
---

## [Editor Report · Acceptance letter]

21 Jul 2021

Dear Dr. Wu,

We are delighted to inform you that your manuscript, " DENV NS1 and MMP-9 cooperate to induce vascular leakage by altering endothelial cell adhesion and tight junction ," has been formally accepted for publication in PLOS Pathogens.

Best regards,

Kasturi Haldar

Editor-in-Chief

PLOS Pathogens

orcid.org/0000-0001-5065-158X

Michael Malim

Editor-in-Chief

PLOS Pathogens

orcid.org/0000-0002-7699-2064